# Cohesin depleted cells rebuild functional nuclear compartments after endomitosis

Marion Cremer [1,11✉], Katharina Brandstetter [2,11], Andreas Maiser [2], Suhas S. P. Rao[3,4], Volker J. Schmid [5], Miguel Guirao-Ortiz [2], Namita Mitra[3], Stefania Mamberti [6], Kyle N. Klein[7], David M. Gilbert [7], Heinrich Leonhardt [2], M. Cristina Cardoso [6], Erez Lieberman Aiden[3,8,9,10], Hartmann Harz [2✉] & Thomas Cremer [1✉]

Cohesin plays an essential role in chromatin loop extrusion, but its impact on a compartmentalized nuclear architecture, linked to nuclear functions, is less well understood. Using live-cell and super-resolved 3D microscopy, here we find that cohesin depletion in a human colon cancer derived cell line results in endomitosis and a single multilobulated nucleus with chromosome territories pervaded by interchromatin channels. Chromosome territories contain chromatin domain clusters with a zonal organization of repressed chromatin domains in the interior and transcriptionally competent domains located at the periphery. These clusters form microscopically defined, active and inactive compartments, which likely correspond to A/B compartments, which are detected with ensemble Hi-C. Splicing speckles are observed nearby within the lining channel system. We further observe that the multilobulated nuclei, despite continuous absence of cohesin, pass through S-phase with typical spatio-temporal patterns of replication domains. Evidence for structural changes of these domains compared to controls suggests that cohesin is required for their full integrity.

[1] Anthropology and Human Genomics, Department Biology II, Ludwig-Maximilians-Universität München, München, Germany. [2] Human Biology & BioImaging, Center for Molecular Biosystems, Department Biology II, Ludwig-Maximilians-Universität München, München, Germany. [3] Center for Genome Architecture, Department of Molecular and Human Genetics, Baylor College of Medicine, Houston, TX, USA. [4] Department of Structural Biology, Stanford University School of Medicine, California, USA. [5] Bayesian Imaging and Spatial Statistics Group, Department of Statistics, Ludwig-Maximilians-Universität München, München, Germany. [6] Cell Biology and Epigenetics, Department of Biology, Technische Universität Darmstadt, Darmstadt, Germany. [7] Department of Biological Science, Florida State University, Tallahassee, FL, USA. [8] Center for Theoretical Biological Physics, Rice University, Houston, TX, USA. [9] Broad Institute of the Massachusetts Institute of Technology and Harvard University, Cambridge, MA, USA. [10] Departments of Computer Science and Computational and Applied Mathematics, Rice University, Houston, TX, USA. [11] These authors contributed equally: Marion Cremer, Katharina Brandstetter. ✉email: Marion.Cremer@lrz.uni-muenchen.de; harz@biologie.uni-muenchen.de; Thomas.Cremer@lrz.uni-muenchen.de

Cohesin, a ring-like protein complex with its major subunits RAD21, SMC1, and SMC3 exerts its key functions by tethering distant genomic loci into chromatin loops. It is involved in sister chromatid entrapment to ensure proper chromosome segregation during mitosis, in double-strand break repair and gene regulation, and importantly was found essential for chromatin loop extrusion by shaping loops in the sub-Mb range anchored at CTCF/cohesin binding sites[1–6], for reviews see[7–13]. These results have argued for an essential role of cohesin in the formation of a functional nuclear architecture.

Studies of the impact of cohesin depletion on nuclear structure and function have become greatly facilitated by an auxin-inducible degron (AID) system, which triggers rapid and selective proteolysis of RAD21 after addition of auxin to the culture medium resulting in the loss of cohesin from chromatin[14]. Using this system in the colon cancer-derived HCT116-RAD21-mAC cell line, we previously demonstrated the rapid disappearance of chromatin loop domains with a concomitant loss of topologically associated domains (TADs) in Hi-C contact matrices averaged over large cell populations, with only minor effects of cohesin depletion on gene expression[15]. Other studies, using different cell types and approaches for cohesin elimination yielded similar results, reviewed in ref. [16].

In this work, we investigate the fate of cohesin depleted cells up to 30 h with both live-cell and super-resolved quantitative microscopy and ensemble Hi-C. These approaches complement each other in ways that cannot be achieved by either method alone. We show that cohesin depleted interphase cells are able to pass through an endomitosis yielding a single postmitotic cell with a multilobulated cell nucleus (MLN). Higher-order chromatin architecture and compartmentalization, typical for cells studied in the presence of cohesin, is maintained after cohesin depletion and even fully restored in MLN as indicated by chromosome territories (CTs), co-aligned active and inactive nuclear compartments (ANC/INC) based on microscopic studies, reviewed in[17,18], as well as the reconstitution of A and B compartments detected by ensemble Hi-C experiments, whereas TADs remain missing. In line with these principal features of a functional nuclear architecture, we find that MLN are able to initiate and traverse through S-phase with typical stage-specific patterns of replication domains (RDs). Quantitative 3D image analyses indicate a larger number of RDs together with an increased heterogeneity of RD volumes. Evidence for structural changes of RDs compared to controls, however, suggests that cohesin is required for their full integrity[19]. A joint presentation of results from quantitative 3D microscopy and Hi-C studies is complicated by a different terminology to describe the structural and functional higher-order chromatin entities discovered by either approach. For a glossary of terms, as we use them below, we refer readers to Supplementary Table 1.

## Results

### Validation of auxin-induced proteolysis of the cohesin subunit RAD21.
All experiments of this study were performed with the human colon cancer-derived cell line HCT116-RAD21-mAC, where an AID is fused to both endogenous RAD21 alleles together with a sequence coding for a fluorescent reporter[14]. About 98% of nuclei in untreated control cell cultures expressed RAD21-mClover. Selective degradation of RAD21 under auxin treatment (6 h in 500 μM auxin) was shown by negative immunostaining with a RAD21 antibody, while epitopes of cohesin subunits SMC1 and SMC3 remained intact under auxin (Supplementary Fig. 1a). RAD21-mClover degradation was quantitatively assessed by intensity measurements recorded from high-throughput imaging of single cells after 6 h auxin treatment (Supplementary Fig. 1b).

A visible decline of RAD21-mClover fluorescence was first noted in time-lapse images 30 min after incubation of cells in 500 μM auxin and appeared completed within 4:00 h (Supplementary Fig. 2a). Furthermore, quantitative measurements of RAD21-mClover decline over time were performed on a single-cell level (for details see Supplementary Fig. 2b, c). Notably, ~4% of cells escaped auxin-induced RAD21 degradation. In order to exclude non-responsive cells from further analyses of the impact of cohesin depletion, RAD21-mClover fluorescence was routinely recorded in all experiments with auxin-treated cell populations except for 3D-FISH experiments where DNA heat denaturation degrades the reporter fluorescence[20].

### Cohesin depleted cells pass through a prolonged endomitosis yielding a daughter cell with one multilobulated nucleus (MLN).
Using time-lapse imaging over 21 h at $\Delta t = 15$ min, we compared entrance into mitosis, mitotic progression and exit in parallel in untreated controls and in cohesin depleted cells, where auxin was added just before starting live-cell observations. In control cells ~80% of all recorded mitoses ($n = 45$) passed mitosis within <1 h and formed two inconspicuous daughter nuclei. A second mitosis observed for individual nuclei ~20 h after the first division demonstrates their capacity to divide again under the given observation conditions (Fig. 1a). Notably, about 20% of mitoses recorded in untreated control cells revealed prolonged mitoses (>2 h) followed by transition into an abnormal cell nucleus (for detailed information on individual nuclei see Supplementary D 1), a feature which is not unusual in tumor cell lines (reviewed in ref. [21]). In cohesin depleted cells ($n = 36$) mitotic entrance was inconspicuous (Fig.1b), mitotic progress, however, was consistently delayed up to 14 h (median 4.5 h, for detailed information on individual nuclei see Supplementary D 1). This prolonged mitotic stage raised the mitotic index in cohesin depleted cell cultures after 6 h in auxin to almost 30% versus ~4% in control cultures (Supplementary Fig. 3). The delayed mitotic passage was associated with the formation of abnormal, e.g., multipolar mitotic figures persisting over several hours (Fig. 1b). Figure 1c depicts a mitotic cell apparently approaching the stage of two separated daughter nuclei. Despite their seemingly complete separation, these daughter nuclei were presumably still connected by filaments (see below and Supplementary Fig. 4) and did not complete karyokinesis. All cohesin depleted cells that were followed through an entire mitosis ($n = 23$, Supplementary D 1) resulted in the formation of a single MLN within one daughter cell, indicative for an endomitotic event[22]. As a consequence, in cell cultures fixed 30 h after cohesin depletion, MLN accumulated up to ~60% versus ~2% in control cultures (Supplementary Fig. 3).

### (Super-resolution) microscopy demonstrates the persistence of global features of higher-order chromatin organization after cohesin depletion and their restoration after endomitosis in MLN.
The capability of cohesin depleted cells to pass through an endomitosis prompted a careful comparison of the architecture of MLN compared with nuclei from control cultures and cohesin depleted cells on their way towards endomitosis (referred to as pre-mitotic cohesin depleted nuclei below). Maintenance of a territorial organization of interphase chromosomes in pre-mitotic, cohesin depleted cells and the reconstitution of chromosome territories (CTs) after endomitosis was demonstrated by the painting of CTs 4, 12, and 19 (Fig. 2). In line with the near-diploid karyotype of HCT116 cells[23], two homologous territories of each painted chromosome were detected in interphase nuclei of both control (Fig. 2a) and pre-mitotic cohesin depleted cells fixed after 6 h in auxin (Fig. 2b). Mitoses occurring in cohesin

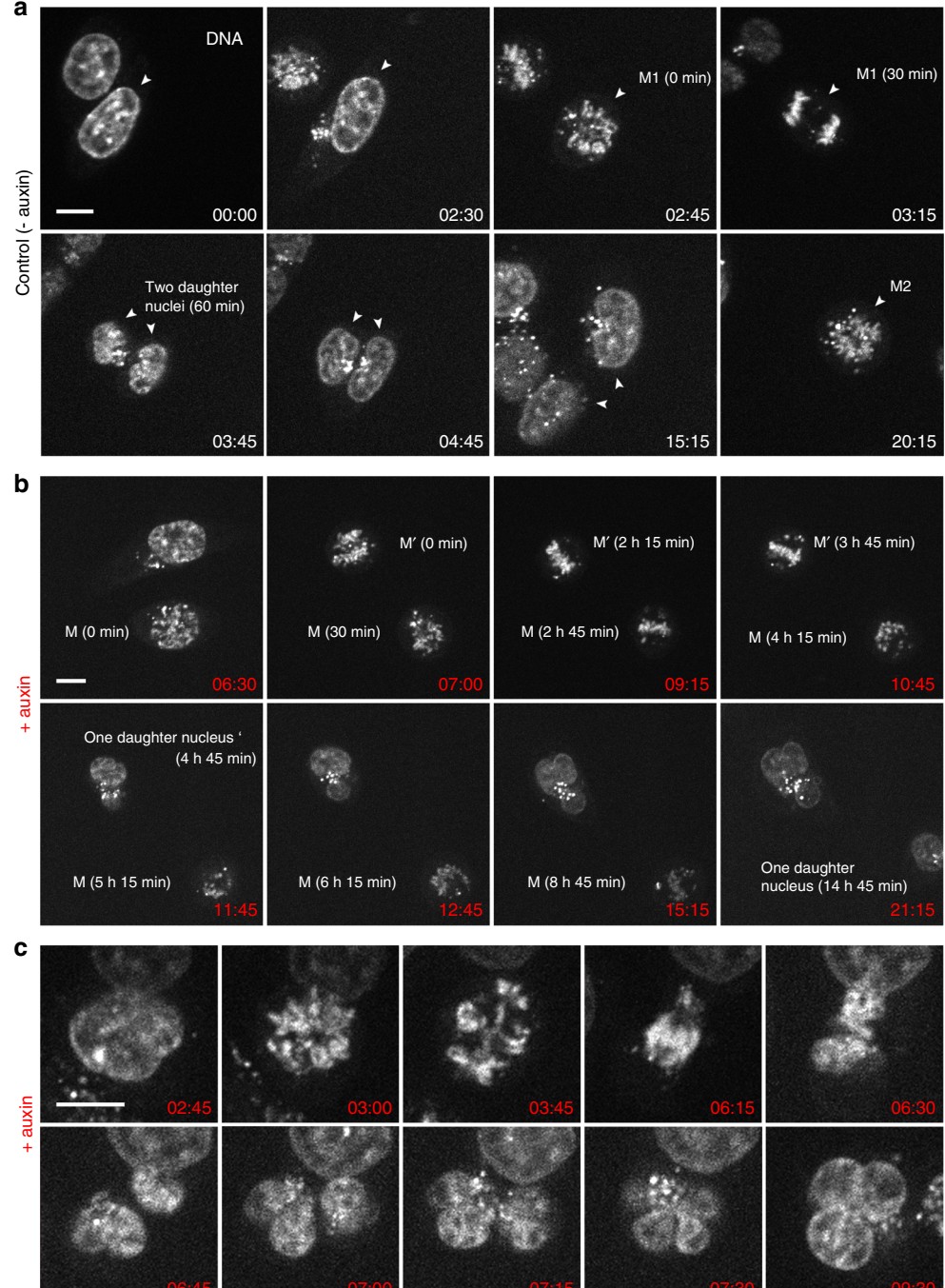

**Fig. 1 Live-cell microscopy demonstrating prolonged abnormal mitosis with subsequent formation of one endomitotic multilobulated nucleus (MLN) in cohesin depleted cells. a** Selected points from time-lapse imaging ($\Sigma t = 21$ h, $\Delta t = 15$ min) of untreated control cells ($-$ auxin) with the accomplishment of mitosis (M1) with in 1h (time 02:45–03:45) and subsequent formation of two daughter nuclei. DNA stained with SiR-DNA. A second mitosis (M2) of one daughter nucleus is shown at time 20:15. **b** Selected time lapse images of nuclei after cohesin degradation ($+$ auxin) conducted in parallel to control cells demonstrate a prolonged mitotic stage. Mitosis (M) emerges at time 6:30 after auxin treatment, transition into one abnormal multilobulated daughter nucleus (MLN) is seen 14:45 h later (time 21:15). Mitosis (M') emerges 7 h after auxin treatment (time 07:00), transition into an MLN is seen 4:45 h later (time 11:45). **c** Time-lapse imaging from the same series at a higher zoom shows an aberrant mitosis with an adumbrated formation of two daughter nuclei (time 06:45), that finally appear as one MLN at time 7:15. Scale bar: 10 μm. M, M1, M2, M' denote different mitoses. Images shown in **a–c** show representative nuclei from one of three independent experiments.

depleted cell cultures observed at this time revealed chromatid segregation, though frequently with misalignment (Fig. 2c). Most MLN fixed in cultures after 30 h of auxin treatment revealed four painted territories for each delineated chromosome (Fig. 2d). Some MLN showed more than four painted regions with variable sizes, which were occasionally connected by thin chromatin bridges (Fig. 2d right panel, Supplementary Fig. 4). These observations may indicate that chromatids were torn apart by mechanic forces during lobe formation. Such disruptions might be enhanced, if we assume a higher level of relaxation and

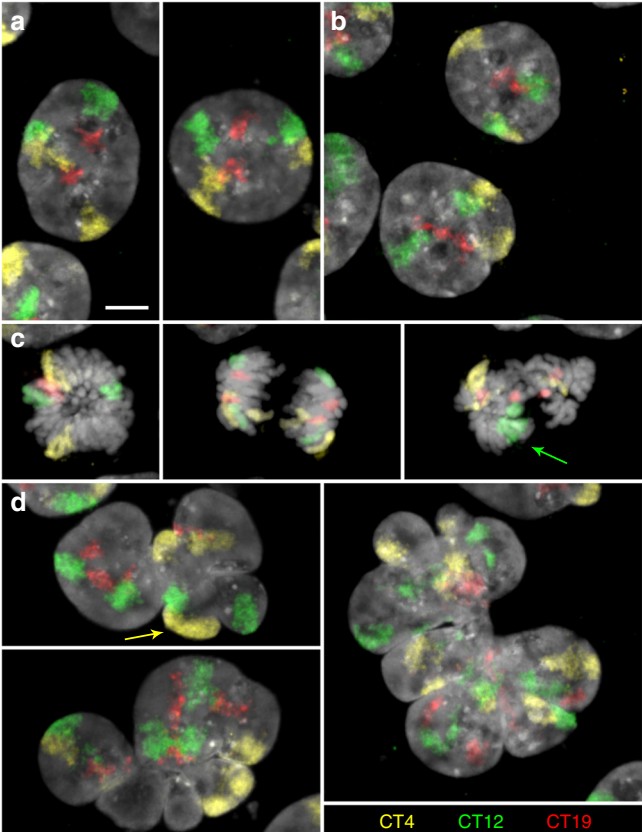

CT4    CT12    CT19

**Fig. 2 Maintenance of chromosome territories (CTs) in cohesin depleted nuclei and their reconstitution after endomitosis. a–d** Z-projections of entire DAPI-stained nuclei (gray) with painted territories of chromosomes 4 (CT4, yellow), 12 (CT12, green) and 19 (CT19, red) acquired by confocal fluorescence microscopy. **a** control nuclei and **b** pre-mitotic cohesin depleted nuclei after 6 h in auxin show two inconspicuous copies for each CT. **c** Mitoses from 6 h auxin-treated cultures with two coherent chromosomes in a (presumably early) metaphase plate (left), after chromatid segregation (mid) and missegregation of chromosome 12 (arrow) in an abnormal mitotic figure (right). **d** left: two endomitotic multilobulated nuclei (MLN) with four copies for each CT. Arrow marks two CTs 4 that are overlayed in the z-projection. Right: Large MLN with a torn-up appearance of CTs with seemingly >4 painted regions for each CT (compare also Supplementary Fig. 4). Scale bar: 5 µm. Images shown in **a–d** show representative nuclei from two independent experiments.

increased mechanical instability of chromosomes in cohesin depleted nuclei.

Next, we tested the ability of cohesin depleted cells to preserve in addition to CTs other structural features of a compartmentalized nuclear architecture with active and inactive nuclear compartments described in the ANC-INC model[17,18]. For this, we compared DAPI-stained nuclei of cohesin depleted cells fixed after 6 h in auxin, mostly comprising nuclei of the pre-mitotic interphase, and post-endomitotic MLN fixed after 30 h auxin treatment with control nuclei of cells cultured without auxin. Functionally relevant markers, delineated by immuno-detection, included SC35, an integral protein of splicing speckles involved in co-transcriptional splicing and transcriptional elongation[24], Ser5P-RNA Pol II, representing a transcription initiating form[25] (further referred to as RNA Pol II), and histone H3K27me3 conveying a repressed chromatin state[26]. 3D structured illumination microscopy (3D-SIM) was used to obtain stacks of nuclear serial sections from representative samples for further evaluation with our previously developed toolbox for 3D image analysis[27].

This toolbox allowed highly resolved measurements of DNA intensity differences as proxies for chromatin compaction combined with the assignment of functional markers to regions of different compaction.

Figure 3a–c shows typical mid-plane SIM sections of a control nucleus (a), a pre-mitotic cohesin depleted nucleus (b), and a post-endomitotic MLN (c). Color-coded voxels were attributed to seven intensity classes with equal intensity variance and represent the range of DAPI fluorescence intensities in 3D SIM nuclear serial sections. These color heat maps visualize local differences in DNA compaction[27]. According to the ANC-INC model (see also Supplementary Table 1 for details of terminology), class 1 represents the interchromatin compartment (IC) with only sparse occurrence of DNA (blue). Chromatin domains (CDs) attributed to classes 2–7 form chromatin domain clusters (CDCs) with a nanoscale zonation of euchromatic and heterochromatic regions[18,28]. Classes 2 and 3 (purple and red) comprise less compacted chromatin, including purple-coded chromatin directly bordering the IC, termed perichromatin region (PR). Classes 4–6 (orange, light brown, yellow) comprise facultative heterochromatin with higher compaction, class 7 (white) reflects the most densely compacted, constitutive heterochromatin. Enlargements of boxed areas in the three mid-plane nuclear sections of Fig. 3a–c exemplify CDCs with a zonal organization of less compact chromatin domains at the periphery adjacent to the IC and higher compacted chromatin located in the CDC interior. Each CT is built from a number of CDCs, which in turn form higher-order chromatin networks expanding throughout the nuclear space, where 3D FISH with appropriate probes is required to identify individual CTs (compare Fig. 2) and CDCs (see Discussion).

Relative fractions of voxels assigned to each of the seven DAPI intensity classes yielded similar patterns for control nuclei, pre-mitotic cohesin depleted nuclei, and post-endomitotic MLN (Fig. 3d). Figure 3e presents estimates of nuclear volumes derived from 3D SIM serial sections. Whereas volumes of pre-mitotic cohesin depleted nuclei are similar to controls, the distinctly increased nuclear volume in MLN (30 h auxin) corresponds with a further increase of a 2n DNA content immediately after endomitosis to a 4n DNA content (Supplementary Fig. 5) after passing through another round of DNA replication (see below). IC-channels expanding between lamina associated chromatin further illustrate the strikingly similar nuclear topography of higher-order chromatin organization present in control nuclei, pre-mitotic cohesin depleted nuclei, and post-endomitotic MLN (Supplementary Fig. 6). 3D image stacks reveal the integration of IC-channels and lacunas into an interconnected 3D network with direct connections to nuclear pores[18,29].

Figure 4a–f shows nuclear sections with DAPI-stained DNA (gray) together with immunostained SC35 (red) and H3K27me3 (green) (Fig. 4a–c) or immunostained RNA Pol II (green) (Fig. 4d–f). In 3D SIM stacks of control and cohesin depleted nuclei we determined the relative fractions of voxels representing SC35, H3K27me3, and RNA Pol II, respectively, in relation to the seven DAPI intensity classes[27]. By comparison of the relative fractions of marker voxels with DAPI related voxels, we tested for each class, whether a given marker showed a relative enrichment (over-representation) or relative depletion (under-representation) compared with the null-hypothesis of a random distribution (Fig. 4g, h). These data present the combined results from two independent experiments (replicates 1 and 2) that were performed with an interval of several months to test their long-term reproducibility. Statistical tests are listed in the source data file. Figure 4g indicates a pronounced enrichment of SC35 in class 1 (IC), a relative depletion in classes 2 and 3 (PR), and a virtual absence in higher classes. In contrast, H3K27me3, a marker of facultative heterochromatin, was under-represented in classes 1 and

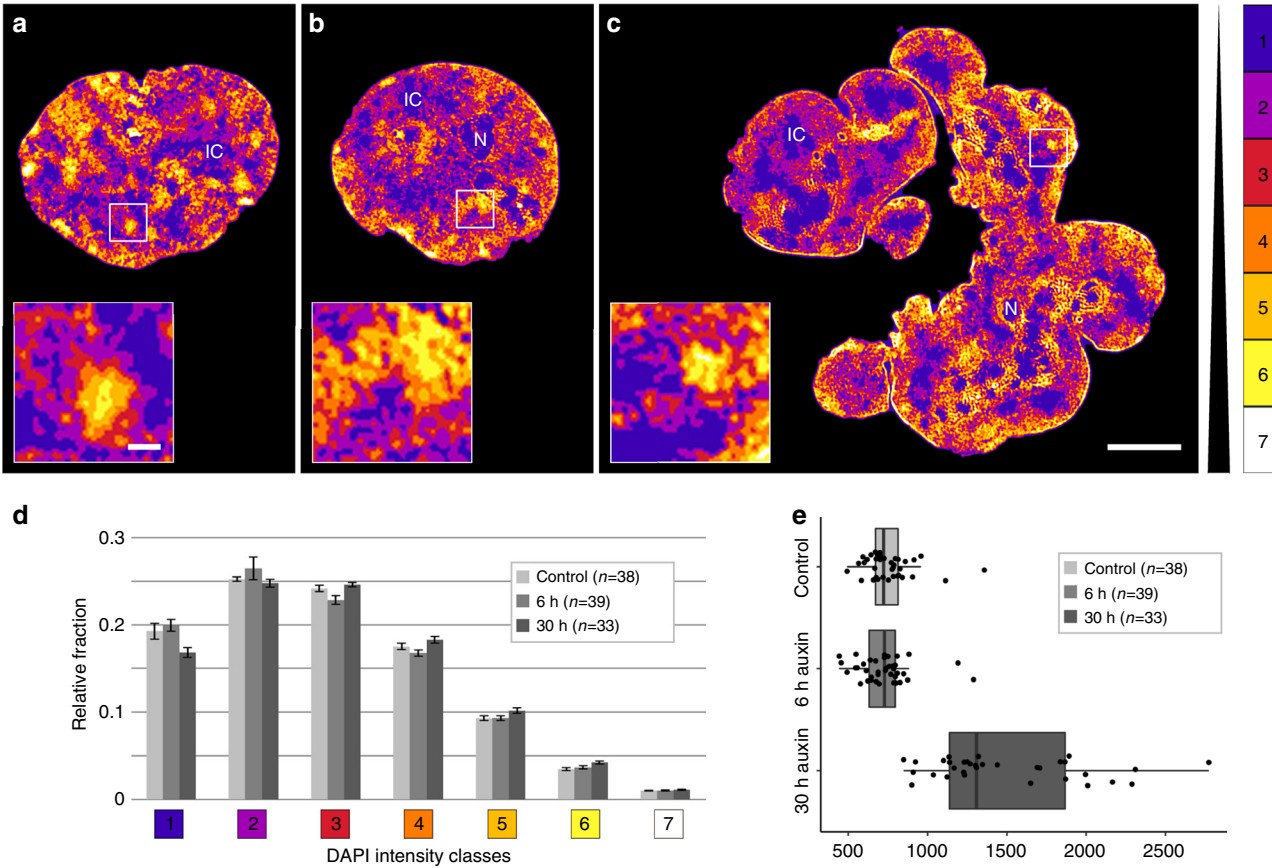

**Fig. 3 Compartmentalized architecture with an interchromatin (IC) channel network pervading chromatin domain clusters (CDC) with zonal compaction differences both in controls and cohesin depleted, pre- and post-endomitotic nuclei. a–c** DAPI-stained mid-sections of representative nuclei acquired by 3D-SIM from **a** control nucleus; **b** cohesin depleted nucleus (6 h auxin); **c** cohesin depleted multilobulated nucleus (MLN) (30 h auxin) are displayed by seven DAPI intensity classes in false colors, used as proxies for chromatin compaction. The color code on the right indicates the assignment of DAPI signals into seven classes with equal intensity variance. This approach allows threshold-independent signal intensity classification based on the intensity of an individual voxel. Class 1 (blue) pixels close to background intensity, largely reflecting the interchromatin compartment (IC) with only sparse DNA, class 7 (white) pixels with the highest intensities. All nuclei in a-c reveal a network of chromatin domain clusters (CDCs) comprising a compacted core and a surrounding low-density zone co-aligned with class 1 regions that meander between CDCs as part of the IC (see insets). Likewise, all nuclei display a rim of compacted (hetero)chromatin at the nuclear periphery. N = nucleolus; IC = interchromatin channels/lacunae. Images in **a–c** show representative nuclei from two independent experiments. Scale bars: 5 μm, insets: 0.5 μm. **d** Relative 3D signal distributions of DAPI intensity classes in control nuclei and cohesin depleted nuclei show an overall similar profile. (control: n = 38, 6 h: n = 39, 30 h: n = 33 cells from two independent experiments). 6 and 30 h, respectively denote incubation time in auxin. Data are represented as mean ± SEM. **e** Average nuclear volumes (μm³) from the same series of nuclei. The ~2-fold increase of nuclear volumes in (post-endomitotic) MLN after 30 h auxin likely reflects their further increase of a 2n DNA content immediately after endomitosis to a 4n DNA content after another round of DNA replication (Supplementary Fig. 5), for statistical tests see Source Data file. Data in **e** are represented as boxplot where the middle line indicates the median, the lower and upper hinges correspond to the 25 and 75% quartiles, the upper whisker extends to the largest value no further than 1.5 × IQR (inter-quartile range) from the hinge and the lower whisker extends to the smallest value from the hinge at most 1.5 × IQR. In addition, all data points are plotted individually. Source data are provided as a Source Data file.

2, but clearly enriched in classes 4 and 5. For RNA Pol II (Fig. 4h) we noted the most pronounced relative enrichment in class 2 and relative depletion in classes 4–7. The separate presentation of both replicates (Supplementary Fig. 7a, b) consistently support an enrichment of SC35 in class 1, and of H3K27me3 in class 4 and 5. The particular enrichment of H3K27me3 in classes 3 and 4 and depletion in class 7 is in line with its assignment as a marker for facultative heterochromatin[30]. Enrichment-depletion patterns of RNA Pol II in the two replicates agree with respect to a general enrichment of RNA Pol II in the ANC (class 1–3), and a depletion within the INC, but differ markedly in quantitative details. Whereas replicate 1 shows a pronounced relative enrichment of this enzyme in class 1 and 2 in line with a relative depletion in classes 3 to 7, replicate 2 shows modest RNA Pol II enrichments in classes 2 and 3, together with relative depletions in classes 5–7, but unexpectedly also in class 1 (IC).

It is important to emphasize that relative enrichments and depletions of epigenetic markers and functional proteins were defined in the 7 DAPI intensity classes. Differences between replicates 1 and 2 that represent snap-shots from the respective experiments may be attributed to unperceived variations of cell culture conditions. Supplementary Fig. 7c–e demonstrates for example a range of compaction differences between SC35 marked speckles in both control and cohesin depleted nuclei. These examples illustrate the cell-to-cell variability of the nuclear landscape, which cannot be captured by a typical one-for-all image. We did not further pursue the question, whether this structural variability reflects functional differences between individual cells in the non-synchronized cell populations studied here.

Notwithstanding these differences, both replicates support our major conclusion: Principal features of a compartmentalized

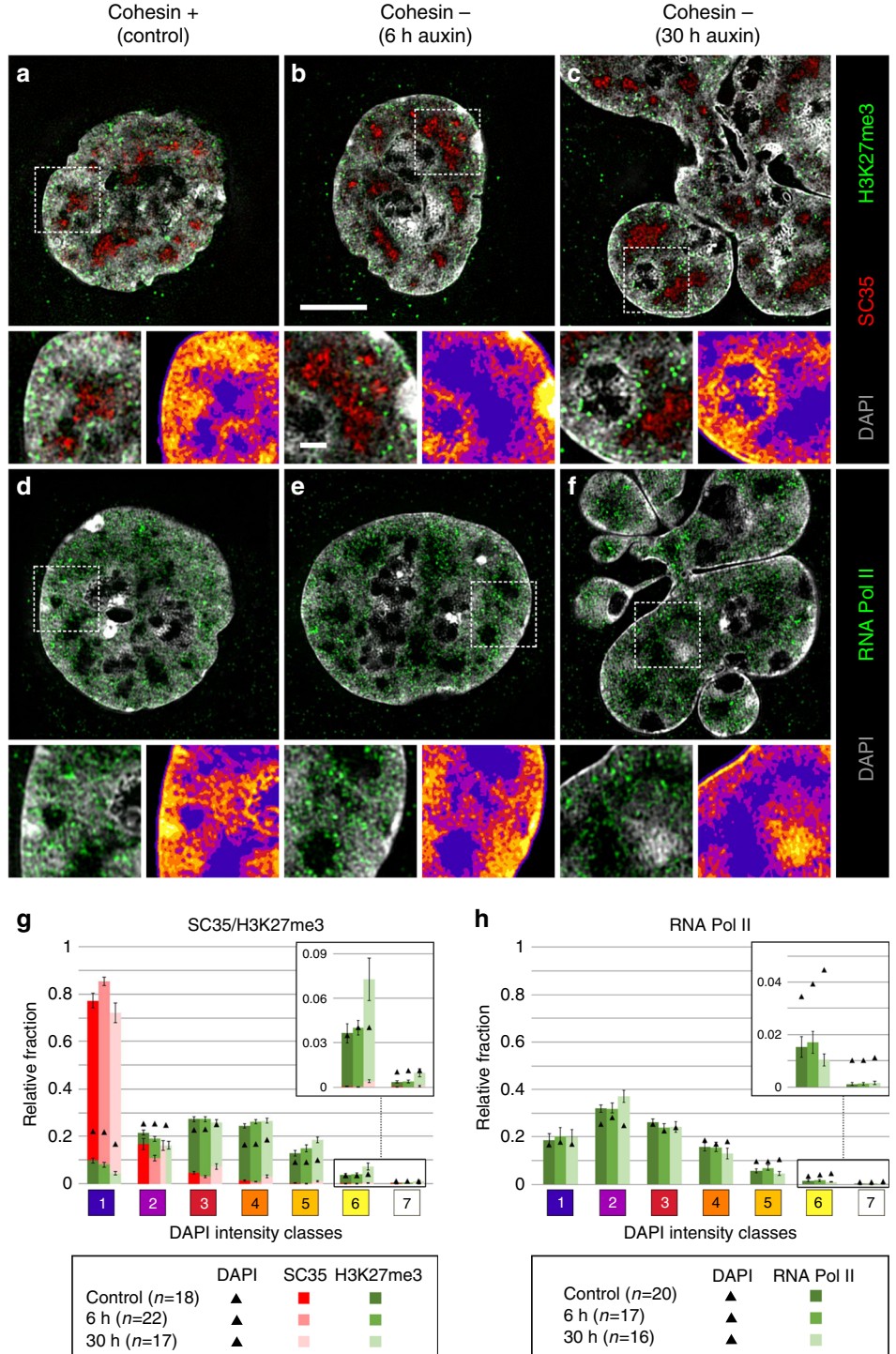

**Fig. 4 Congruent 3D topography of SC35, H3K27me3, and RNA Pol II in nuclei of control and cohesin depleted cells. a–f** DAPI-stained nuclear mid-sections (gray) displayed from 3D SIM image stacks of control nuclei (**a**, **d**), of pre-mitotic cohesin depleted nuclei after 6 h auxin treatment (**b**, **e**), and post-endomitotic MLN after 30 h auxin treatment (**c**, **f**) reveal the topography of immunostained SC35 (red) and H3K27me3 (green) (**a–c**), and active RNA Pol II (red) (**d–f**). Scale bar: 5 µm. An enlargement of a representative boxed area is shown beneath each mid-section together with the color-coded DAPI intensity heat map (compare Fig. 3). Scale bar: 1 µm. SC35 marked splicing speckles are located in the interchromatin compartment (IC, blue), H3K27me3 marks are distributed within neighboring chromatin domain clusters; RNA Pol II is mainly enriched in chromatin lining the IC (purple), but also extends into the IC, whereas it is largely excluded from densely compacted chromatin regions (brown and yellow). **g**, **h** 3D image analyses of 3D SIM stacks show the relative fraction of **g** SC35 (red) and H3K27me3 (green) signals (control: $n = 18$, 6 h: $n = 22$, 30 h: $n = 17$ cells from two independent experiments), and **h** of active RNA Pol II (green) (control: $n = 20$, 6 h: $n = 17$, 30 h: $n = 16$ cells from two independent experiments) in comparison to DAPI intensity classes 1–7 marked as black triangles. 6 and 30 h, respectively, denote incubation time in auxin. Data are represented as mean ± SEM. Source data are provided as a Source Data file.

organization with CTs and CDCs, pervaded by the IC in control nuclei were maintained in pre-mitotic, cohesin depleted nuclei and were rebuilt in post-endomitotic MLN, where individual macromolecules may penetrate into highly compacted CDs while macromolecular aggregates, such as a transcription machinery (RNA Pol II) or splicing machinery (SC35) may be excluded[17,31].

**In situ Hi-C data indicate the maintenance/rebuilding of A and B compartments in cohesin depleted pre-mitotic nuclei and post-endomitotic MLN despite the loss of loop domains.** In situ Hi-C of cell cultures, treated with auxin for 6 and 28 h, respectively, prior to fixation, confirmed the disappearance of loop domains (Fig. 5a) in contrast to control cultures, whereas A and B compartments were maintained (Fig. 5b; for terminology see Supplementary Table 1b). Since most cells had passed an endomitosis with the formation of MLN after 28–30 h auxin treatment (Supplementary Fig. 3), we conclude that these findings are representative for both cohesin depleted pre-mitotic nuclei and post-endomitotic MLN. A heightened compartmentalization was noted in particular with regard to B-type chromatin, as previously described for pre-mitotic cohesin depleted cells[15]. Strengthened interactions between this B-type compartment could be readily observed even in our low depth data from 28 h auxin-treated cells (Fig. 5c, lower right panel, interactions between loci annotated in yellow). While the functional identity or significance of this particular B-type subcompartment remains unknown, we were able to identify by k-means clustering of histone modification data for HCT116-RAD21-mAC cells[15] a histone modification cluster (consisting of depletion of both activating marks like H3K36me3 and H3K27ac and repressive marks such as H3K27me3 and H3K9me3, but a mild enrichment of H3K79me2) that corresponded to the positions of this particular B-type subcompartment (Fig. 5d, e; cluster 4). Genome-wide analysis of the average Hi-C contact frequencies between the histone modification clusters demonstrated a strong enrichment for within-cluster contacts for this B-type subcompartment at both 6 and 28 h after cohesin degradation, and additionally, at 28 h, mild cohesin-degradation induced enrichment of interactions between this B-type subcompartment and clusters enriched for repressive histone modifications as well as depletion of interactions with clusters enriched for activating histone modifications. The comparison of ensemble Hi-C data with microscopic data described above supports the argument that A/B compartments and ANC/INC compartments reflect the same structures (see Discussion).

**Persistence of typical S-phase stage replication patterns after cohesin depletion.** The following part of our study shows that the structural compartmentalization of pre-mitotic, cohesin depleted cells and post-endomitotic MLN corresponds with their functional capability to maintain RDs and to proceed through S-phase. The temporal order of replication is highly coupled with genome architecture, resulting in typical patterns for early, mid, and late replication timing[32]. RDs were chosen in our study as microscopically visible reference structures, which correspond to microscopically defined chromatin domains (CDs) and persist as stable chromatin entities throughout interphase and during subsequent cell cycles[33–35] (Supplementary Table 1a). Replicating DNA was visualized by pulse replication labeling (RL) (see Methods). Control cultures were fixed 6 h after RL (Fig. 6a), cultures prepared for cohesin depletion were further grown after RL for 1 h under normal medium conditions and then exposed to auxin for 6 h (Fig. 6b, c) or 30 h (Fig. 6d) before fixation. Both controls (a) and auxin-treated cells (b, d) revealed nuclei with typical RD patterns for different S-phase stages. This experiment demonstrates that different RD patterns persist during the

subsequent pre-mitotic interphase of cohesin depleted cells (Fig. 6b) and can be fully reconstituted in post-endomitotic MLN (Fig. 6d). Notably, structural entities reflecting RDs pulse-labeled during S-phase can also be identified along mitotic chromosomes (Fig. 6c). MLN are able to initiate a new S-phase with the formation of typical replication patterns, shown in Fig. 6e, where RL was performed in cultures after 30 h of auxin treatment (Fig. 6e).

**Replication timing is preserved upon cohesin depletion.** Using Repli-Seq[36,37] and Hi-C analysis, replication timing was measured by the ratio of early to late replicating DNA and was found preserved upon cohesin depletion (Fig. 7a, b), consistent with a prior report[38]. In addition, the tight relationship between genome A/B compartmentalization and replication timing was similarly maintained in the absence of cohesin, exemplified for chr. 8 (Fig. 7a). Data were based on at least two replicates of each timepoint and confirmed the reproducibility of results.

**Structural changes of RDs in cohesin depleted nuclei.** Finally, we tested whether cohesin depletion results in structural changes of individual RDs, detectable on the resolution level of 3D-SIM (Fig. 8). For this purpose, RD counts and RD volumes were evaluated in nuclei of three cultures: The control culture was fixed 6 h after RL together with the 6 h auxin culture, which was incubated with auxin immediately after RL. The 30 h auxin culture was fixed after 30 h in auxin, when most cells had passed an endomitosis yielding a multilobulated cell nucleus. Nuclei with RD patterns typical for early S-phase at the time of pulse labeling were identified in the three fixed cultures and 3D serial image stacks of such nuclei were recorded with SIM and used for measurements in entire nuclei. It is important to note that an RD pattern generated by pulse labeling in a given nucleus is maintained after S-phase and after mitosis, independent of the time of fixation during the post-endomitotic interphase of MLN. Therefore, controls and auxin-depleted cells fixed 6 h after RL proceeded to G2, but still showed the early S-phase RD pattern. In the culture fixed 30 h in auxin, we identified MLN also showing early replication patterns. Figure 8a shows examples of such nuclei from the control culture (left), the 6 h cohesin depleted culture (middle) and from 30 h cohesin depleted MLN cells (right). Figure 8b presents average numbers of segmented RDs for individual nuclei and Fig. 8c shows the results of volume estimates for individual RDs from the respective nuclei. Compared with controls, we noted an increase of both RD numbers and volumes together with an increase of heterogeneity (broader range of number and size distribution) in cohesin depleted pre-mitotic nuclei and post-endomitotic MLN. Based on the concordant increase of counts and volumes of segmented RDs in cohesin depleted nuclei in comparison with control nuclei, we tentatively conclude that cohesin is indispensable to prevent disintegration and decompaction of RDs (see Discussion).

**Effect of cohesin depletion on DNA halo induced chromatin loops.** An effect of cohesin depletion on chromatin loop structure was supported by a DNA halo approach, a technique to investigate changes in chromatin organization at the level of DNA loops[39]. Histone extraction in interphase nuclei by high-salt incubation triggers the extrusion of chromatin loops from a densely stained central chromatin core thus providing a measure of their size. DAPI-stained nuclei of cohesin depleted cells (6 h auxin treatment) exhibited halos that were significantly larger and more variable in shape in comparison to the defined and compacted halos of control cells (Supplementary Fig. 8) in line with the recently described observation that the cohesin-NIPBL complex compacts DNA by extruding DNA loops[19].

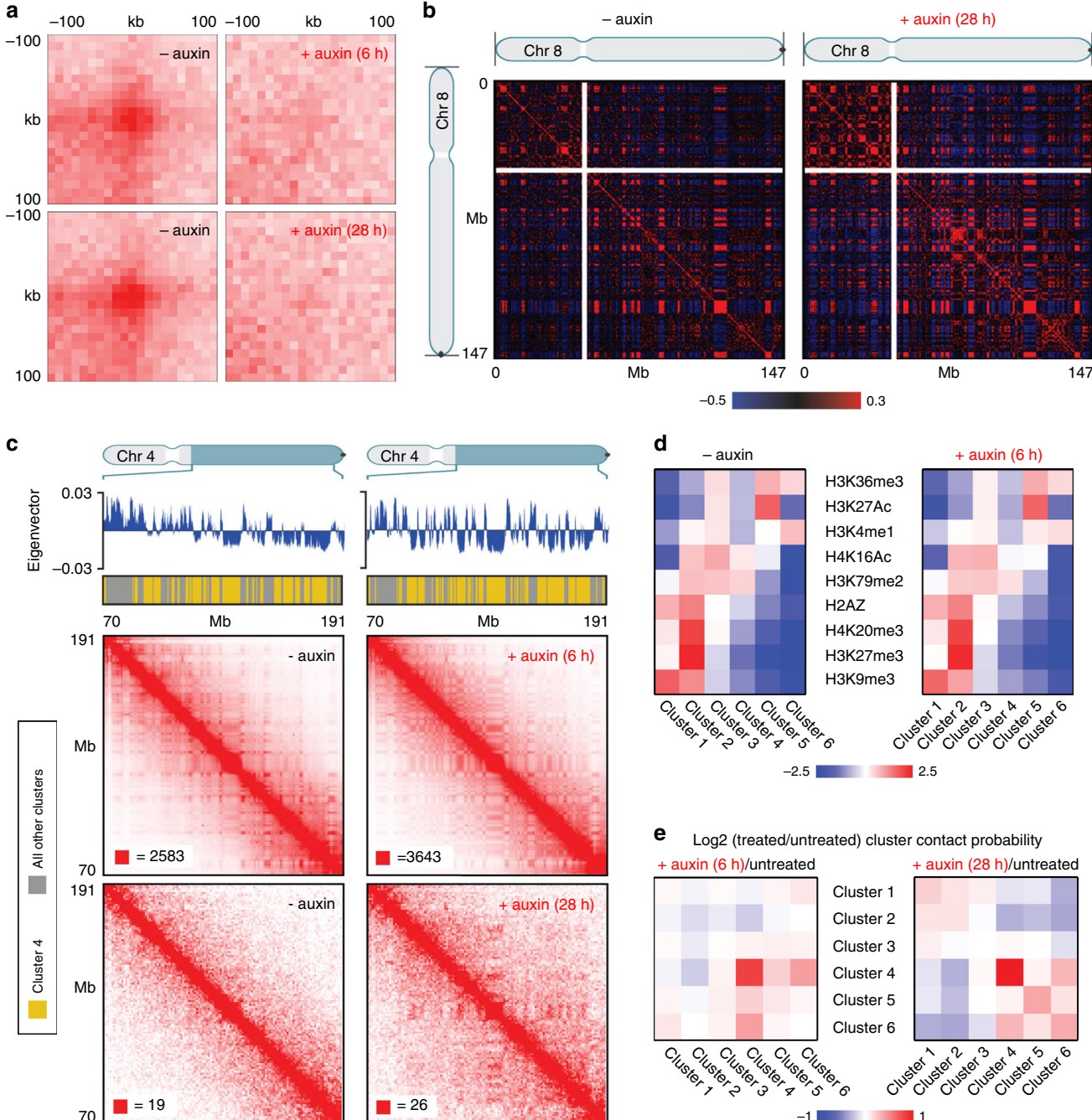

**Fig. 5 Hi-C data indicate elimination of chromatin loops, but the maintenance of A and B compartments in cohesin depleted pre- and postmitotic cells.**
**a** Aggregate peak analysis (APA) plots using loops identified in HCT116-RAD21-mAC cells before and after 6 h of auxin treatment (top) or before and after 28 h of auxin treatment (bottom). For each of the treated timepoints, the matched untreated control (harvested at the same time) is plotted next to it. The plot displays the total number of contacts that lie within the entire putative peak set at the center of the matrix. Loop strength is indicated by the extent of focal enrichment at the center of the plot. **b** Pearson's correlation maps at 500 kb resolution for chromosome 8 before (left) and after (right) 28 h of auxin treatment. The plaid pattern in the Pearson's map, indicating compartmentalization, is preserved in cohesin depleted nuclei even after 28 h of auxin treatment. **c** Contact matrices for chromosome 4 between 70 and 191 Mb at 500 kb resolution before (left) and after (right) cohesin depletion. The 6 h cohesin depletion time is shown on top, and 28 h depletion time on the bottom. K-means clustering of histone modifications at 25 kb resolution into six clusters annotates loci corresponding to specific subcompartments. Interactions for loci in cluster 4 (arbitrary numbering, annotated in yellow on top tracks) are strengthened after both 6 and 28 h of cohesin depletion. All loci belonging to clusters other than cluster 4 are annotated in gray in the top track. The max color threshold (red) of the heatmap is illustrated in the lower-left corner of each heatmap, the minimum color threshold (white) is 0 reads. **d** Log-2 fold ratios of between-cluster Hi-C contact probabilities post- and pre- cohesin degradation are shown for the six clusters identified via K-means clustering of histone modifications. Cluster 4 shows strong contact probability enrichment after cohesin degradation at both the 6 h and 28 h timepoints. **e** For each of the six histone modification clusters, the average log2-fold enrichment for each histone modification over all loci in that cluster is shown both post- and pre- cohesin degradation. Patterns of histone modifications across the clusters as unchanged by cohesin degradation.

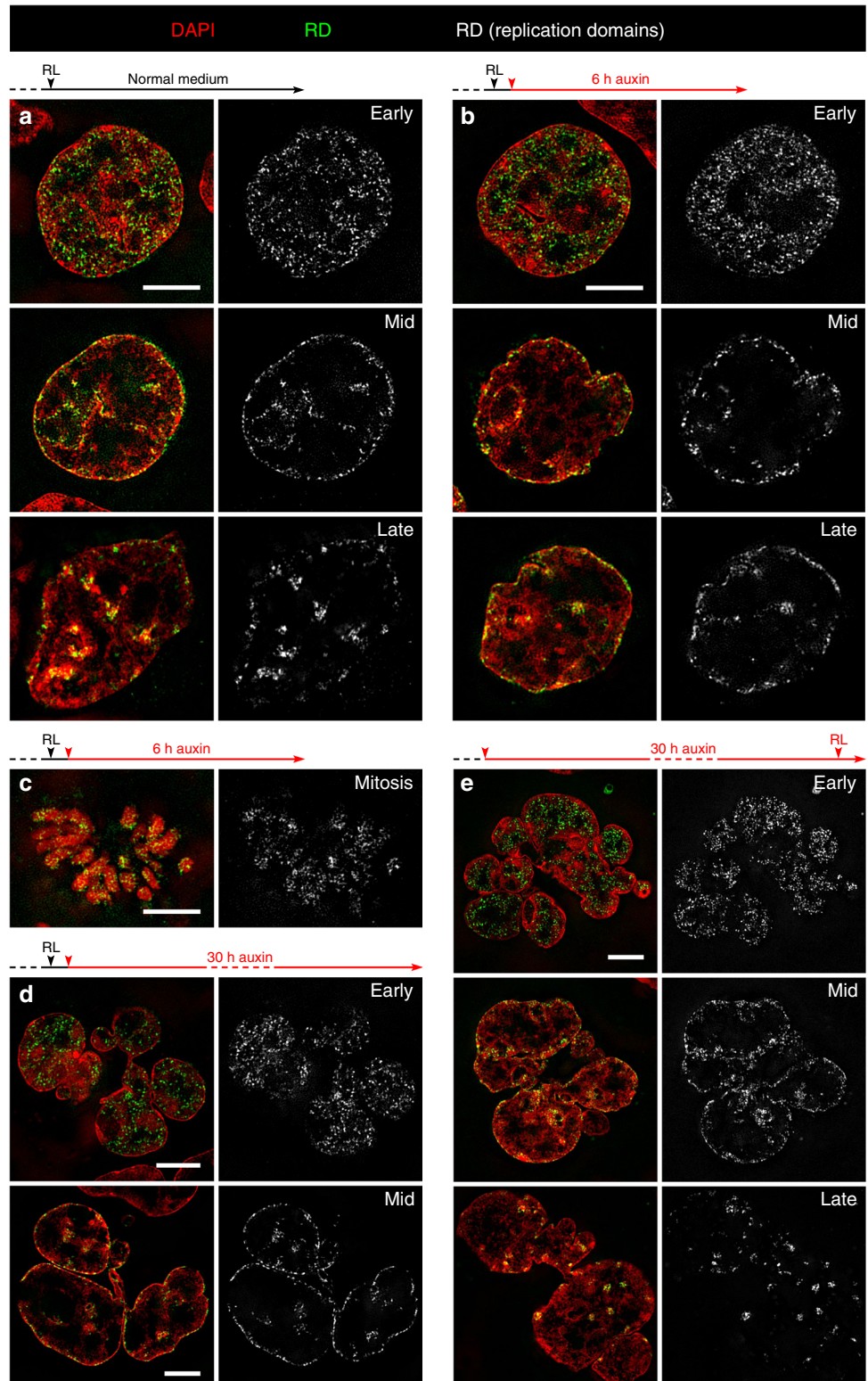

**Fig. 6 Maintenance, postmitotic rebuilding and de novo formation of typical replication patterns after cohesin depletion. a–e** Overlay images (left) show representative SIM sections of DAPI-stained nuclei (red) with replication domains (RDs)(green) identified by replication labeling (RL) in different stages of S-phase. RDs in the same nuclear sections are also displayed in gray (right). **a** Control nuclei fixed 6 h after RL with typical patterns for early, mid and late replication, respectively. **b** Maintenance of the same typical replication patterns in nuclei of cohesin depleted, pre-mitotic cells fixed 6 h after RL. **c** Cohesin depleted mitotic cell with replication labeled chromatin domains obtained under conditions as described in **b**. **d** RD patterns in individual lobuli demonstrate the ability of post-endomitotic MLN to restore RD patterns, generated by RL during the previous cell cycle. Cells were treated with auxin for 30 h after RL. **e** RL carried out with MLN obtained after ~30 h auxin treatment demonstrates de novo DNA synthesis with the formation of new typical replication patterns. Scale bar: 5 µm. Images shown in **a**–**e** show representative nuclei from three independent experiments.

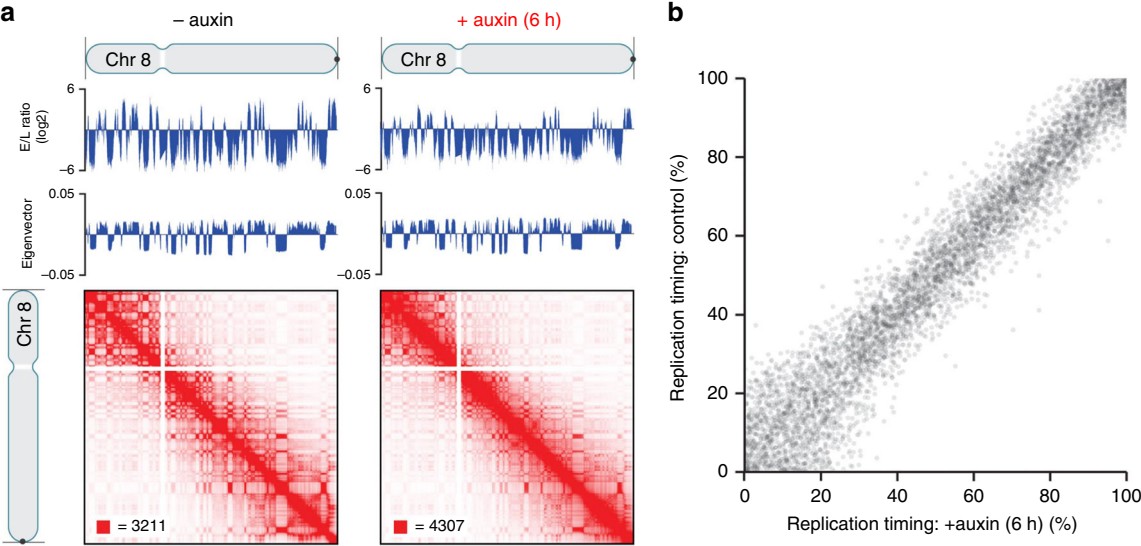

**Fig. 7 Hi-C and Repli-Seq data demonstrate the same replication timing for cohesin depleted and non-depleted control cells. a** Contact matrices of chromosome 8 at 500 kb resolution along with the corresponding Repli-Seq early-to-late (E/L) ratio tracks at 50 kb resolution and the first eigenvectors of the Hi-C matrices corresponding to A/B compartmentalization. Replication timing along the genome is conserved, as shown by the correspondence of the untreated and auxin-treated Repli-Seq tracks. In addition, the correspondence between replication timing and genome compartmentalization (as indicated by the plaid pattern in the Hi-C map and the first eigenvector of the Hi-C matrices) is preserved after auxin treatment. **b** Scatter plot of replication timing (percentile of E/L ratio) in RAD21-mAC cells before (*y*-axis) and after (*x*-axis) auxin treatment.

## Discussion

Our study demonstrates that multilobulated nuclei (MLN), that arise from cohesin depleted cells after passing through an endomitosis, retain the ability to rebuild a compartmentalized nuclear architecture. Whereas ensemble Hi-C confirmed the continued absence of chromatin loops and TADs in MLN as in pre-mitotic cohesin depleted nuclei, A and B compartments were fully restored in MLN in line with active and inactive nuclear compartments (ANC and INC[17,18]) revealed by 3D SIM. In light of the fundamental roles ascribed to cohesin, the capacity of MLN to initiate another round of DNA replication with stage-specific patterns of RDs was not expected.

Progression of cells into a disturbed and prolonged mitosis after cohesin depletion by Rad21 siRNA transfection was described in previous live-cell studies covering ~4 h[40]. By extending the live-cell observation period up to 21 h, we discovered a so far unreported endomitosis with chromatid segregation, but apparent failure to complete karyokinesis and cytokinesis. This failure may be attributed to the impact of cohesin for proper spindle pole formation and kinetochore-microtubule attachment (reviewed in refs. [7,8]). Notably, in vertebrates loading of cohesin onto DNA already occurs in telophase[7,41], which may be essential for subsequent cytokinesis and daughter cell formation. Factors promoting endomitosis and the formation of MLN are, however, complex and certainly diverse[42]. Multipolar endomitosis with the formation of polyploid MLN occurs physiologically in megakaryocytes[43] and in (cohesin competent) tumor cell lines[44], in part entailing extensive chromosomal rearrangements[45]. The observation of MLN as the mitotic outcome in ~2% of HCT116-RAD21-mAC control cells exemplifies the spontaneous occurrence of MLN in a near-diploid tumor cell line.

Hi-C and related methods offer the great advantage of a genome-wide approach to explore a nuclear compartmentalization at the DNA sequence level. This approach demonstrated a compartmentalized architecture of the landscape in cohesin depleted cell nuclei[4,16], but failed to detect the profound global morphological changes in post-endomitotic cohesin depleted

MLN compared to cohesin depleted nuclei before passing through endomitosis. High-resolution microscopy is also the method of choice to examine the 3D structure of chromatin domain clusters (CDCs) with a zonal organization of repressed (condensed) and transcriptionally competent (decondensed) chromatin domains and the actual 3D configuration of the interchromatin compartment (IC)[46] with its supposed function as storage and transport system[17] that co-evolved with higher-order chromatin organization[47]. Our results exemplify the necessity to combine bottom-up with top-down approaches in ongoing 4D nucleome research, aimed at a comprehensive understanding of the structure-function relationships in complex biological systems.

We propose that microscopically defined ANC/INC compartments and A/B compartments, detected by ensemble Hi-C, represent the same functional compartments. Chromatin that contributes to the ANC and compartment A, respectively, is gene rich, transcriptionally active, and typically located preferentially in the interior of mammalian cell nuclei, whereas both the INC and compartment B comprise gene poor, transcriptionally repressed chromatin of higher compaction, which is more prominent at the nuclear periphery (for review see refs. [18,48]). We further propose to equate microscopically defined chromatin domains (CDs)/RDs comprising several hundred kb (see below) that constitute functional building blocks of the ANC and INC with similarly sized compartment domains (see Supplementary Table 1c) as functional building blocks of A and B compartments rather than with TADs[49–51]. A correspondence of microscopically discernible RDs with TADs mapped by ensemble Hi-C has been favored in some studies[50,51].

TADs represent genomic regions between several 100 kb up to >1 Mb in length, where DNA sequences physically interact with each other more frequently compared to sequences outside a given TAD[48,52–54]. TADs, however, do not represent an individual chromatin structure, but a statistical feature of a cell population. Boundaries detected in Hi-C experiments are noted as transition points between TAD-triangles. They constrain, but do not restrict completely the operating range of regulatory

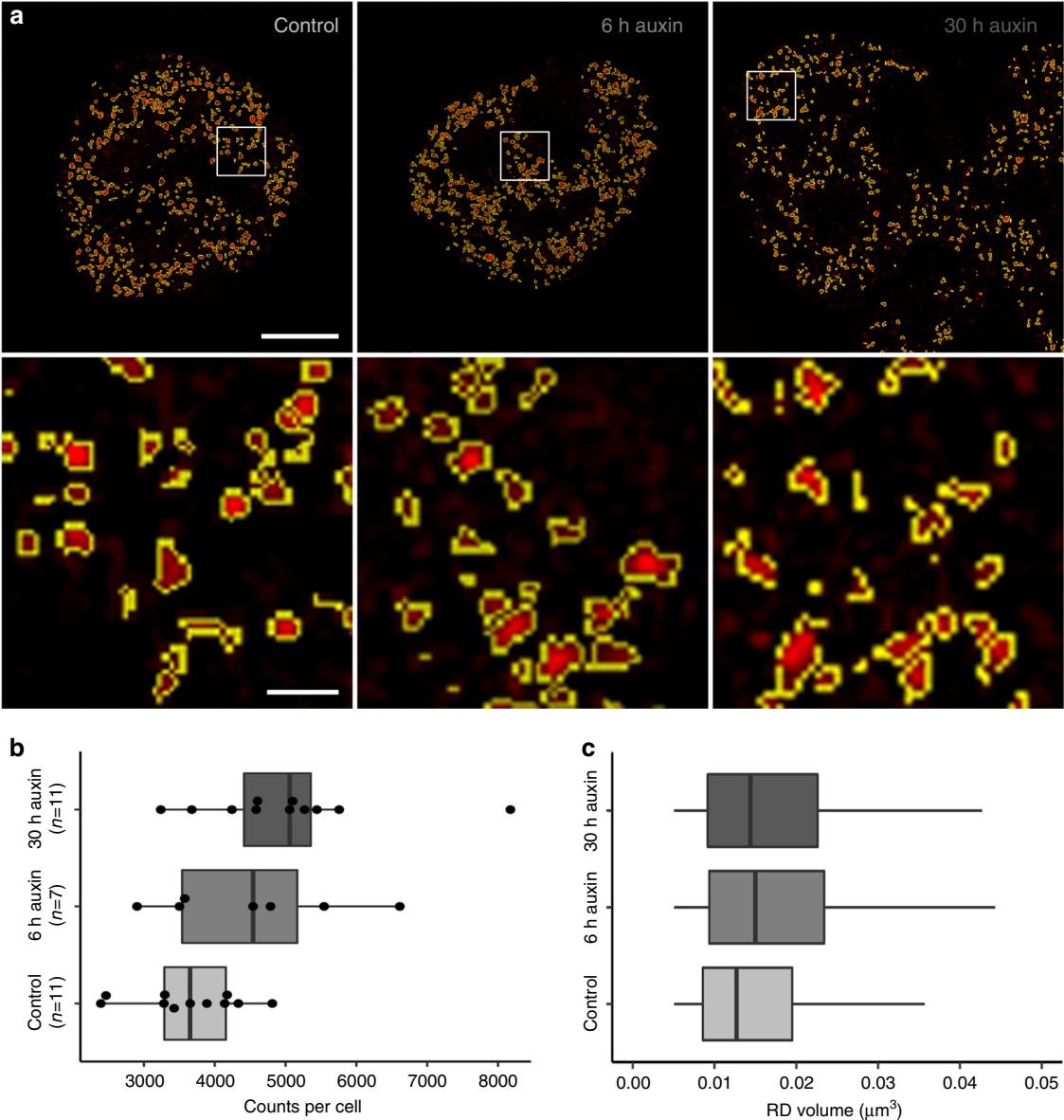

**Fig. 8 Segmentation of individual replication domains (RDs), pulse-labeled in early S-phase, indicate structural changes after cohesin depletion. a** SIM nuclear mid-sections of nuclei with typical early S-phase patterns of replication domains (RDs) from a control culture (left), a 6 h auxin-treated culture (middle), both fixed 6 h after RL, and a multilobulated nucleus (right) obtained after 30 h auxin treatment (compare Fig. 6a–d). Enlargements of boxed areas show individual, segmented RDs displayed in red with segmented borders lined in yellow. Scale bar: 4 μm, 0.5 μm in inset magnifications. **b** Counts of segmented RDs plotted for control nuclei, cohesin depleted nuclei after 6 h auxin and MLN after 30 h auxin are presented as dots (control: $n = 11$, 6 h: $n = 7$, 30 h: $n = 11$ nuclei from two independent experiments). **c** Boxplots with corresponding volume distributions of segmented individual RDs (control: $n = 39.334$, 6 h: $n = 31.467$, 30 h: $n = 55.153$). The non-parametric two-sided Mann–Whitney test revealed significant differences of RD counts between control nuclei and MLN ($p = 0.012$) and for RD volumes ($p < 0.0001$ for control<-> 6 h auxin, control<-> 30 h auxin, and 6 h auxin<-> 30 h auxin). Volumes of RDs with dimensions below the resolution limit of 3D-SIM (~120 nm lateral/300 nm axial) show the same size and were excluded from consideration. Accordingly, the lower limits of volumes between control nuclei and cohesin depleted nuclei are identical in contrast to the differences of the upper volume limits. Data in **b**, **c** are represented as boxplots where the middle line indicates the median, the lower and upper hinges correspond to the 25 and 75% quartiles, the upper whisker extends to the largest value no further than 1.5 × IQR (inter-quartile range) from the hinge and the lower whisker extends to the smallest value from the hinge at most 1.5 × IQR. In **b**, all data points are plotted individually. Source data are provided as a Source Data file.

sequences[55]. Recently, super-resolution microscopy demonstrated the presence of TAD-like domains at the single-cell level[56]. In cohesin depleted cells, a more stochastic placement of borders between TAD-like domains was detected[57]. A role of IC-channels as additional structural boundaries between CDs and CDCs located on both sides, has been considered but not proven[17].

Early microscopic studies of the replicating genome during S-phase provided a first opportunity to explore its genome-wide partitioning into discrete structural entities with a DNA content of ~1 Mb, called RDs or replication foci[58,59]. We adopted the term ~1 Mb chromatin domains in line with evidence that RDs persist as similarly sized stable chromatin units throughout interphase and during subsequent cell cycles[33,34]. Later studies assigned an average DNA content of 400–800 kb to RDs/CDs[37], which can be optically resolved down to clusters of a few single replicons (150–200 kb)[35,60]. Gene rich, early replicating domains

form the A compartment, gene poor, later replicating domains the B compartment[37].

Our study confirms previous reports, which showed the maintenance of pulse-labeled RDs and the formation of S-phase specific replication patterns in cohesin depleted, pre-mitotic interphase cells[28,38]. In addition, our study demonstrates the ability to re-constitute RDs in a typical pattern arrangement in post-endomitotic MLN. Moreover, MLN were able to initiate a new round of DNA replication with the formation of typical stage-specific replication patterns under continued absence of cohesin.

These observations, however, do not imply that cohesin would be dispensable for RD structure. A comparison of numbers (counts) and volumes of individual RDs generated in early S-phase in nuclei of control cells and cells treated with auxin for 6 and 30 h revealed a significant increase both of RD numbers and RD volumes and also in a remarkably increased heterogeneity of these parameters after cohesin depletion. The near double amount of RD numbers in MLN (30 h auxin) compared to controls was expected since MLN are generated as a result of an endomitosis with full separation of sister chromatids harboring RDs where labeled nucleotides were incorporated into both newly synthesized DNA strands in the previous cell cycle. In cohesin depleted cells treated with auxin after RL for 6 h the increase of discernible RDs may result from an enhanced untethering of labeled sister chromatids compared to controls. At the time of fixation, both controls and cohesin depleted cells had likely reached the late S or G2-phase and labeled RDs had formed two separate sister chromatids within a given CT. Sister chromatids are kept together by cohesin at some sites, but are untethered at other sites and can dissociate from each other up to few hundred nm[61]. In cohesin depleted nuclei these untethered sites are likely increased. An increase of RD counts based on RD splitting should correspond with a decrease of RD volumes. Unexpectedly, we observed a remarkable volume increase in individual segmented RDs. This observation supports a role of cohesin in the compaction of chromatin structures exerted by chromatin loop extrusion[19] which could affect contact frequencies and thus explain at least in part the loss of TAD patterns in ensemble Hi-C experiments. Due to the resolution limit of 3D-SIM (~120 nm lateral/300 nm axial) these results must be viewed with caution: a fraction of RDs with sizes below this limit would show a putative size reflecting the diffraction limit, resulting in an overestimate of their volumes. To overcome these method-inherent limitations, imaging approaches with higher resolution, such as STORM/SMLM or STED are required to further clarify the influence of cohesin on RD structure[61,62]. The increased heterogeneity of RD volumes in cohesin depleted nuclei compared with controls, likely reflects the cell-to-cell shift of boundaries described for TAD-like domains in cohesin depleted cells[57]. In summary, the current study supports our previous conclusion[15] that cohesin plays an indispensable role for the structure of RDs/CDs, but is dispensable for the formation of a compartmentalized nuclear organization with preserved A and B compartments. These earlier results are substantially enhanced through the microscopic observations described in the present study, which demonstrates that cohesin depleted cells passing through an endomitosis are able to rebuild a cell nucleus with the basic features of the ANC and INC, respectively. It is of note to emphasize here that Hi-C alone did neither allow to detect the drastic changes of the global architecture between cohesin depleted cells studied before and after endomitosis nor the added topographical features of IC-channels and lacunas. The current study may help to stimulate integrated research strategies with the goal to better understand the structure-function implications of the nuclear landscape.

New methods of super-resolved optical reconstruction of chromatin organization with oligopaints technology[56] or the combination of serial block-face scanning electron microscopy with in situ hybridization (3D-EMISH)[63] have opened up new ways to explore the geometrical variability of TAD-like structures in comparison with TADs identified by ensemble Hi-C and to close current gaps of knowledge on nuclear compartmentalization. Despite compelling evidence for chromatin loops, their actual 3D and 4D (space-time) organization is not known. Microscopic evidence for the formation of higher-order chromatin arrangements based on nucleosome clutches or nanodomains[28,56,57,64–66] suggests that loops may be organized as much more compact structures with the potential implication that the diffusion of individual macromolecules into their interior may be constrained and the penetration of macromolecular aggregates is fully excluded[31]. As a consequence, transcription and other nuclear functions may preferentially occur at the surface of chromatin clusters, dynamically remodeled to fulfill this condition. How dynamic changes of functionally defined higher-order chromatin structures in space and time are related to changing functional requirements of cells at different levels of a hierarchical chromatin organization, defines major challenges for future studies. Such studies should also advance our still incomplete knowledge of cohesin functions.

## Methods

**Cells and culture conditions.** HCT116-RAD21-mAID-mClover cells (referred to as HCT116-RAD21-mAC cells in the manuscript) were generated and kindly provided by the Kanemaki lab (Mishima Shizuoka, Japan). For a detailed description see ref. [14]. Cells were cultured in McCoy's 5A medium supplemented with 10% FBS, 2 mM L-glutamine, 100 U/ml penicillin, and 100 μg/ml streptomycin at 37 °C in 5% $CO_2$. For data shown in Supplementary Fig. 1b and 8 HCT116-RAD21-mAC cells and HCT-116 wild-type cells were grown in in DMEM medium supplemented with 10% FCS, 2 mM L-glutamine, 50 μg/ml gentamicin. Cells were tested for mycoplasma contamination by confocal microscopy.

**Auxin-induced RAD21 proteolysis.** Degradation of AID-tagged RAD21 was induced by the addition of auxin (indole-3-acetic acid; IAA, Sigma Aldrich) to the medium at a final concentration of 500 μM (auxin stock solution 2 M in DMSO). In long-term cultures fresh auxin-medium was added after ~18 h.

**Immunodetection.** Immunodetection was performed on cells grown to 60–80% confluency on high precision borosilicate glass coverslips (170 μm ± 5 μm thickness) with the following antibodies: cohesin subunits RAD21 (Abcam), SMC1 and SMC3 (both Bethyl laboratories) were detected with Cy3-conjugated goat anti rabbit antibodies (Dianova). Primary antibodies against SC35 (Sigma), RNA Pol II (Abcam), and H3K27me3 (Active Motif) were detected with either donkey anti-mouse Alexa 488 (Life technologies) or donkey-anti rabbit Alexa 594 (Life technologies). To meet the requirements for super-resolution microscopy with respect to an optimal signal-to-noise ratio and preservation of 3D chromatin structure, a protocol described in[67] was followed: Cells were washed with 1×PBS (pH = 7.4 w/o Ca/Mg) and fixed in 4% paraformaldehyde / PBS for 10 min. During the last 3 min the fixative was stepwise replaced by PBS/0.02% Tween followed by 2 × 5 min washing with PBS/0.02% Tween. The samples were quenched with 20 mM glycine/PBS for 10 min, washed with PBS/Tween, permeabilized with 0.5% Triton X-100/PBS for 10 min and subsequently incubated in blocking buffer (150 mM NaCl, 15 mM Hepes/KOH, 2 mM $MgCl_2$, 0.1 mM EGTA, 0.2% Triton X-100, 0.5% fish skin gelatin, 2% BSA) for 1.5–2 h. Antibodies were diluted in blocking buffer and incubated for 1–1.5 h (primary) or 45–60 min (secondary) followed by intensive washing with blocking buffer. Cells were postfixed in 4% paraformaldehyde/PBS for 5 min, washed in PBS/Tween and counterstained in 2 μg/ml DAPI (4′,6-diamidino-2-phenylindole) for 10 min, washed in PBS. Cells were mounted in antifade mounting medium (Vectashield (Vector Laboratories)) and the cover slips sealed with nail varnish onto a conventional microscopic slide. For the specification of antibodies see Supplementary Table 2.

**Replication pulse labeling (RL) by replication scratch labeling[34].** Cells cultivated on high precision coverslips (thickness 0.170 mm) grown to 50–80% confluency were transferred into a dry empty tissue dish after draining off excess medium. Thirty microliters of a prewarmed labeling solution containing 20 μM Cy3-dUTP (homemade) or Alexa 594-5-dUTP (Life technologies) was evenly distributed over the coverslip. With the tip of a hypodermic needle parallel scratches at distances of ~100 μm were quickly applied to the cell layer. Cells were

incubated for 1 min in the incubator, then a few ml of pre-warmed medium was added to the dish. After 30 min medium was exchanged to remove non-incorporated nucleotides. This procedure preserves the RAD21-mClover fluorescence after labeling.

**RL by incorporation of 5-Ethynyl-dU (EdU) and detection via click chemistry.** This approach was used for RL in MLN after 30 h auxin treatment (compare Fig. 6e) since these cells are prone to detachment upon scratching. EdU was added at a final concentration of 10 μM to the medium for 15 min. Incorporated EdU was detected according to manufactures instructions (baseclick) by a Cu(I) catalyzed cycloaddition reaction that covalently attaches a fluorescent dye containing a reactive azide group to the ethynyl-group of the nucleotide[68]. For visualization of RDs, the dye 6-FAM-Azide (baseclick) at a final concentration of 20 μM was used.

After either labeling approach cells were washed in 1×PBS, fixed with 4% formaldehyde/PBS for 10 min and permeabilized with 0.5% Triton X-100/PBS/Tween 0.02% for 10 min. Cells were counterstained in 1 μg/ml DAPI and mounted in antifade mounting medium (Vectashield (Vector Laboratories); for details, see ref. [67]).

**Hi-C in situ analysis of untreated and auxin-treated cells.** HCT-116-RAD21-mAC cells were plated in 6-well plates with either complete media, or complete media with 500 μM auxin (IAA) for 6 h (as in ref. [15]) or 28 h (to enrich for post-mitotic cells with MLN). Cells were crosslinked with 1% formaldehyde directly on the plate for 10 min and then quenched with glycine. The crosslinked cells were then scraped off and in situ Hi-C was performed. In brief, cells were permeabilized with nuclei intact, the DNA was digested overnight with MboI, the 5′-overhangs were filled in while incorporating bio-dUTP, and the resulting blunt end fragments were ligated together. Crosslinks were then reversed overnight, the DNA was sheared to 300–500 bp for Illumina sequencing, biotinylated ligation junctions were captured using streptavidin beads and then prepped for Illumina sequencing. We prepared three libraries (two biological replicates) each for each time point (untreated 6 h, treated 6 h, untreated 28 h, treated 28 h). All Hi-C data were processed using Juicer[69,70]. The data were aligned against the hg19 reference genome. All contact matrices used for further analysis were KR-normalized with Juicer. Comparison of compartment strengthening to histone modification clusters was done as in ref. [15]. Histone modification data for 9 marks (H3K36me3, H3K27Ac, H3K4me1, H4K16Ac, H3K79me2, H2AZ, H4K20me3, H3K27me3, and H3K9me3) generated from untreated and 6-hour treated cells in[15] was grouped into six clusters using k-means clustering. For the k-means clustering, the histone modification data were first converted into a z-score value for each mark in order to account for differences in the dynamic range between marks.

**Repli-Seq of untreated or auxin-treated cells.** HCT116-RAD21-mAC cells were synchronized in G1 with lovastatin following a protocol of ref. [71]. Cells were incubated with 20 μM Lovastatin (Mevinolin) (LKT Laboratories M1687) for 24 h to synchronize in G1. Five-hundred micromolar auxin or DMSO was added 6 h before release from lovastatin block. To release from G1 block, lovastatin was washed away with three washes of PBS and warm media plus 2 mM Mevalonic acid (Sigma-Aldrich M4667) and 500 μM auxin or DMSO. Cells were released for 10, 14, 18, and 22 h. Two hours before the time point 100 μM BrdU was added to label nascent replication. After fixation, equal numbers of cells from each release time point were pooled together for early/late repli-seq processing[36]. Repli-Seq data were processed as described in ref. [36]. In brief, data were aligned to the hg19 reference genome using bowtie2, deduplicated with samtools, and the log-2 ratio between early and late timepoints was calculated.

**3D DNA-FISH.** Labeled chromosome painting probes delineating human chromosomes 4-(BIO), 12-(DIG) and 19-Cy3 were used. Following the protocol described in ref. [20], 30 ng of each labeled probe and a 20-fold excess of COT-1 DNA was dissolved per 1 μl hybridization mix (50% formamide/2 × SSC/10% dextran sulfate). Cells were fixed with 4% formaldehyde/PBS for 10 min. After a stepwise exchange with 0.5% Triton X-100/PBS, cells were permeabilized with 0.5% Triton X-100/PBS for 10 min. Further pretreatment steps included incubation in 20% glycerol (1 h), several freezing/thawing steps in liquid N2, incubation in 0.1 N HCl (5 min) and subsequent storage in 50% formamide/2 × SSC overnight. After simultaneous denaturation of probe and cells (2 min at 76 °C), hybridization was performed at 37 °C for 48 h. After stringent washing in 0.1 × SSC at 60 °C, biotin was detected by streptavidin-Alexa 488 and DIG by a mouse-anti-DIG antibody conjugated to Cy5. Cells were counterstained in 1 μg/ml DAPI, and mounted in antifade mounting medium Vectashield (Vector Laboratories).

**DNA halo preparation.** Cells were incubated for 6 h in 500 μM auxin for cohesin depletion. DNA halo preparation was largely performed according to ref. [72]. After washing the cells in 1xPBS they were incubated for 10 min in a buffer at 4 °C containing 10 mM Tris pH 8, 3 mM MgCl2, 0.1 M NaCl, 0.3 M sucrose, protease inhibitors (freshly added to the buffer prior to use) 1 μM pepstatin A, 10 μM E64, 1 mM AEBSF and 0.5% Nonidet P40. All the following procedures were performed at room temperature. Subsequently, DNA was stained for 4 min with 2 μg/ml DAPI. After 1 min in a second extraction buffer (25 mM Tris pH 8, 0.5 M NaCl,

0.2 mM MgCl2; protease inhibitors as in nuclei buffer and 1 mM PMSF were added fresh prior to use), cells were incubated 4 min in halo buffer (10 mM Tris pH 8, 2 M NaCl, 10 mM EDTA; protease inhibitors as in nuclei buffer and 1 mM DTT were added fresh prior to use). Finally, cells were washed 1 min each in two washing buffers (25 mM Tris pH 8, 0.2 mM MgCl2; the first buffer with and the second without 0.2 M NaCl). After 10 min fixation in 4% formaldehyde/PBS, cells were washed twice in 1×PBS and mounted on slides with Vectashield. Nuclear scaffolds and the faded DNA halos were imaged at a widefield microscope (Zeiss Axioplan 2, 100x/1.30 NA Plan-Neofluar Oil Ph3 objective; Axiovision software (version 4.8.2.0 SP3); AxioCam mRM camera). Both the total area (At) and the scaffold area (As) of each cell were manually segmented using the software Fiji and the DNA halo area (Ah) calculated as a subtraction of the two (Ah = At – As). The DNA halo radius was subsequently derived with the formula $R = \sqrt{(Ah/\pi)}$. Four biological replicates were prepared and measured. For generation of plots and statistical analysis (Wilcoxon test) the software RStudio (version 1.0.143) was used.

**Confocal fluorescence microscopy.** Confocal images were collected using a Leica SP8 confocal microscope equipped with a 405 nm excitation laser and a white light laser in combination with an acousto-optical beam splitter (AOBS). The used confocal system has three different detectors, one photomultiplier tube (PMT) and two hybrid photodetectors (HyD). The microscope was controlled by software from Leica (Leica Application Suite X, ver. 3.5.2.18963). For excitation of DAPI, the 405 nm laser was used, for excitation of Alexa488, Cy3, STAR635P, and Cy5, the white light laser was set to 499, 554, 633, and 649 nm, respectively. The emission signal of DAPI was collected by the PMT (412–512 nm), the emission signals of Alexa488 (506–558 nm), Cy3 (561–661 nm), STAR635P (640–750 nm), and Cy5 (656–780 nm) were collected by the two HyD detectors. Images were acquired with 42 nm pixel steps, 102 μs pixel dwell time and twofold line accumulation using a Leica HC PL APO 63x/1.30 NA Glycerol immersion objective. The frame size was 37 × 37 μm and the scan speed was 700 Hz. The size of the confocal pinhole was 1 A.U. Confocal image z-stacks were acquired with a step size of 330 nm.

**Live-cell microscopy for long-term observations.** For live-cell imaging, cells were plated on poly-L-Lysine-coated glass-bottom 2-well imaging slides (ibidi), allowing to image control and auxin-treated conditions in parallel. For DNA staining cells were incubated in media containing 500 nM SiR-DNA (Spirochrome) for 1 h before imaging. Timelapse acquisitions were carried out on a Nikon TiE microscope equipped with a Yokogawa CSU-W1 spinning disk confocal unit (50 μm pinhole size), an Andor Borealis illumination unit, Andor ALC600 laser beam combiner (405 nm/488 nm/561 nm/640 nm) and Andor IXON 888 Ultra EMCCD camera. The microscope was controlled by software from Nikon (NIS Elements, ver. 5.02.00). Cells were imaged in an environmental chamber maintained at 37 °C with 5% CO2 (Oko Labs), using a Nikon PlanApo 60x/1.49 NA oil immersion objective and a Perfect Focus System (Nikon). Images were recorded every 15 min for 21 h as z-stacks with two planes and a step size of 6 μm, unbinned and with a pixel size of 217 nm. For excitation of mClover and SiR-DNA, the 488 and 640 nm laser lines were used, respectively. Fiji software (ImageJ 1.51j)[73] was used to analyze images.

**Quantitation of auxin-induced RAD21-mAID-mClover degradation on single cells after fixation.** HCT-116-RAD21-mAC and HCT-116 wild-type cells were treated with 500 μM auxin for 6 h, fixed in 3.7% formaldehyde, permeabilized with 0.7% Triton X-100 for 15 min, counterstained with 1 μg/ml DAPI for 10 min and mounted in Vectashield mounting medium (Vector Laboratories). High-throughput imaging of single cells was performed at the wide-field microscope Operetta (40x/0.95 NA air objective; Harmony software (version 3.5.1); Jenoptik firecamj203 camera, Perkin Elmer). The high-content images were analyzed on batch through a pipeline created with the Harmony software and nuclei identified based on DAPI signal. The nuclei found on the border of each field were removed and the remaining nuclei were selected based on morphology parameters, such as size and roundness. mClover intensities were then measured within the nuclear mask of the selected nuclei. The fluorescence intensities data were exported into tables and processed in RStudio (version 1.0.143) to produce plots and statistical analysis. For each treatment, the measurements were combined from three biological experiments, each made of two technical replicates. mClover intensities measured from HCT-116 wild-type cells were used as an estimate for the background level. A median of 10 A. U. (arbitrary units) was calculated for the nuclear mClover intensity in wild-type cells (10.23 and 10.56 A. U. in the untreated and in the auxin-treated wild-type cells, respectively). This background value was subtracted from all values measured for the untreated and auxin-treated HCT-116-RAD21-mAC cells.

**Quantitation of auxin-induced RAD21-mAID-mClover degradation on single cells in time-lapse acquisitions.** Nikon spinning disk confocal live-cell time lapses were acquired as described above. For the analysis the lower of the two planes showing interphase cells was used. The detailed description of segmentation and analysis scripts can be found as comments in the scripts which are deposited on GitHub (https://github.com/CALM-LMU/Cohesin_project.git). In brief, segmentation maps for nuclei in the SiR-DNA channel in confocal time lapses were

obtained by a machine learning-based pixel classification using Ilastik (version 1.3.3) (standard settings). Segmentation maps were manually curated in order to analyze only individual nuclei. Nuclei were traced starting at time frame 1 until the cell entered mitosis and disappeared from the lower imaging plane. The generated segmentation maps were used to select single nuclei in the mClover channel. After background subtraction (modal gray value) the median intensity was measured for each labeled cell over time using Fiji (version 1.51j). Only cells with a mClover intensity above 50 counts were included in the analysis. All data shown are normalized to their starting values. Cells surpassing a fluctuation above the 90% quantile relative to their own rolling mean of 5 timepoints were filtered out. Plots were generated using Python (version 3.7.1).

**DNA content assessment in individual nuclei by integrated DAPI intensity measurement**. DAPI-stained nuclei were acquired using the Nikon spinning disk system described above. Fixed samples of untreated control cells and cells treated with auxin for 30 h were acquired as confocal image z-stacks in 35 planes with a step size of 300 nm using a Nikon PlanApo 100x/1.45 NA oil immersion objective. DAPI was excited with the 405 nm laser line. Segmentation and analysis scripts are described in detail in the scripts which are uploaded on GitHub (https://github.com/CALM-LMU/Cohesin_project.git). In brief, spinning disk confocal stacks of DAPI-stained nuclei were used for a machine learning-based pixel classification to obtain 3D segmentation maps of nuclei using Ilastik (version 1.3.3) (standard settings). Segmentation maps were manually curated in order to analyze only individual non touching nuclei. After background subtraction (modal gray value) the integrated intensity was measured for each segmented DAPI-stained nucleus by using Fiji (version 1.51j). Plots were generated using R Studio (version 1.0.143).

**Semi-automatic quantification of MLN and mitoses**. Image acquisitions were carried out on the Nikon spinning disk system described above. Using a Nikon PlanApo 100x/1.45 NA oil immersion objective tiled images (3 × 3 with 5% overlap and 131 nm pixel size) were acquired for each condition to increase the number of cells per field of view. Confocal image z-stacks were acquired in two planes with a step size of 6 μm in order to encompass cells, in particular mitotic cells, in different plane levels. DAPI and mClover were excited with 405 or 488 nm laser lines, respectively. All nuclei from each image (average 280 nuclei per image frame) were classified visually into morphologically normal nuclei, mitoses, and MLN. In auxin-treated cells nuclei with persistent RAD21-mClover fluorescence (~2%) were excluded.

**Structured illumination microscopy (SIM)**. Super-resolution structured illumination imaging was performed on a DeltaVision OMX V3 system (Applied Precision Imaging/GE Healthcare) equipped with a 100x/1.4 NA UPlan S Apo oil immersion objective (Olympus), Cascade II:512 EMCCD cameras (Photometrics) and 405, 488, and 593 nm lasers (for detailed description, see ref. [74]). For sample acquisition oil with a refractive index of RI = 1.512 was used. 3D image stacks were acquired with 15 raw images per plane (five phases, three angles) and an axial distance of 125 nm using DeltaVisionOMX (version 2.25, Applied Precision Imaging/GE Healthcare) and then computationally reconstructed (Wiener filter setting of 0.002, channel-specific optical transfer functions (OTFs)) and color shift corrected using the SoftWoRx software (version 5.1.0, Applied Precision Imaging/GE Healthcare). After establishing 32-bit composite tiff stacks with a custom-made macro in Fiji/ImageJ2 (http://rsb.info.nih.gov/ij/), the data were subsequently aligned again to get a higher alignment precision. These images were then used for measurements in the Volocity software (version 6.1.2., Perkin Elmer).

**Nuclear volume measurements**. Volume measurements were done with the Volocity software (Version 6.1.2.). RGB image stacks were separated in their respective channels and then nuclei structures were obtained and segmented for volume measurements by using the following commands: (1) "Find Objects" (Threshold using: Automatic, Minimum object size: 200 μm³), (2) "Dilate" (number of iterations: 15), (3) "Fill Holes in Objects" and (4) "Erode" (number of iterations: 15). In ≈5% of cases these settings had to be adjusted for the challenging task of nuclei segmentation. To confirm statistical significance of volume differences the Mann-Whitney test was applied.

**Segmentation and quantification of RD signals**. Aligned 3D SIM image stacks were used as RGB for object counting and volume measurements in the Volocity software (Version 6.1.2.). For each series between $n = 7$ and $n = 11$ nuclei were measured resulting in 31,000–55,000 single values for each series. Image stacks were separated into their respective channels. The segmentation of RD structures was performed with the following software commands: (1) "Find Objects" (Threshold using: Intensity, Lower: 32, Upper: 255), (2) "Separate Touching Objects" (Object size guide of 0.002 μm³) and (3) "Exclude Objects by Size", excluding structures <0.005 μm³. This cut-off level largely corresponds to the resolution limit of 3D-SIM (~120 nm lateral/300 nm axial). Exclusion of signals outside a selected nucleus was achieved by the commands "Intersect" and "Compartmentalize". Segmentation of nuclei was realized by the following commands: (1) "Find Objects" (Threshold using: Intensity), (2) "Dilate", (3)

"Fill Holes in Objects" and (4) "Erode". Measured values of individual object counts and segmented RD volumes were displayed as boxplots indicating the median with 25–75% quartiles. Plots were generated using R Studio (version 1.0.143).

**3D assessment of DAPI intensity classes as proxy for chromatin compaction classification**. Nuclei voxels were identified automatically from the DAPI channel intensities using Gaussian filtering and automatic threshold setting. For chromatin quantification a 3D mask was generated in ImageJ to define the nuclear space considered for the segmentation of DAPI signals into seven classes with equal intensity variance by a previously described in house algorithm[27], available on request. In brief, a hidden Markov random field model classification was used, combining a finite Gaussian mixture model with a spatial model (Potts model), implemented in the statistics software R[75]. This approach allows threshold-independent signal intensity classification at the voxel level, based on the intensity of an individual voxel. Color or gray value heat maps of the seven intensity classes in individual nuclei were performed in ImageJ.

**Quantitative allocation of defined nuclear targets on 3D chromatin compaction classes**. Individual voxels of fluorescent signals of the respective marker channels were segmented by a semi-automatic thresholding algorithm (accessible in VJ Schmid (2020). nucim: Nucleome Imaging Toolbox. R package version 1.0.9. https://bioimaginggroup.github.io/nucim/). XYZ-coordinates of segmented voxels were mapped to the seven DNA intensity classes. The relative frequency of intensity weighted signals mapped on each DAPI intensity class was used to calculate the relative distribution of signals over chromatin classes. For each studied nucleus the total number of voxels counted for each intensity class and the total number of voxels identified for the respective fluorescent signals for SC35, RNA Pol II, H3K27me3 was set to 1. As an estimate of over/under representations (relative depletion/enrichment) of marker signals in the respective intensity classes, we calculated the difference between the percentage points obtained for the fraction of voxels for a given DAPI intensity class and the corresponding fraction of voxels calculated for the respective signals. Calculations were performed on single-cell level and average values over all nuclei were used for evaluation and plotting. For a detailed description, see ref. [27].

**Statistics and reproducibility**. Microscopic observations were verified from at least two independent series. Images shown in the figures are representative images from respective experiments. For highly elaborate quantitative 3D analyses of super-resolved image stacks we selected between 7 and 39 nuclei for a given experiment with the precondition of a high staining and structure-preserving quality. No statistical method was used to predetermine sample size. Investigators were not blinded during the experiments and when assessing the outcome. Significance levels were tested by a non-parametric two-sided Wilcoxon test and a Bonferroni-Holm correction was used to avoid errors through multiple testing when applicable (see Source D). Data shown in column graphs represent mean ± standard error of the mean (SEM), as indicated in the figure legends. The variance was similar between the groups that were statistically compared.

**Reporting summary**. Further information on research design is available in the Nature Research Reporting Summary linked to this article.

## Data availability
Raw and processed Hi-C and Repli-Seq data generated as part of this study can be publicly accessed with NCBI GEO accession GSE145099. Publicly available ChIP-Seq data used in this study are available at GEO accession GSE104888. Raw microscopy data used for Figs. 1–4, 6, 8, Supplementary Figs. 1–8, additional "biological replicates" and complementary experiments can be accessed under https://doi.org/10.5061/dryad.vt4b8gtqb, Cohesin depleted cells rebuild functional nuclear compartments after endomitosis, Dryad, Dataset. All other relevant data supporting the key findings of this study are available within the article and its Supplementary Information files or from the corresponding authors upon reasonable request. Source data are provided with this paper.

## Code availability
Code used in this study for 3D assessment of DAPI intensity classes (Fig. 3) and quantitative allocation of defined nuclear targets on 3D chromatin compaction classes (Fig. 4 and Supplementary Fig. 7) is available under https://bioimaginggroup.github.io/nucim/ and was published in ref. [27]. The website provides a detailed installation guide. Custom Python (version 3.7.1) and R (version 1.0.143) scripts for quantification of RAD21 decay and DNA content analysis (Supplementary Figs. 2 and 5) are available under https://github.com/CALM-LMU/Cohesin_project.git and current versions are provided as Supplementary Software. All Hi-C data were processed using the software package Juicer version 1.5.7, which can be found at https://github.com/aidenlab/juicer. Previously published ChIP-Seq data from ref. [15] were clustered using the scipy.cluster.vq. kmeans function. Repli-Seq data were processed and analyzed exactly following the code published in ref. [36].

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

## Acknowledgements

We thank Stefan Müller, University of Munich (LMU), for kindly providing chromosome painting probes for 3D-FISH experiments and Irina Solovei, University of Munich (LMU), for generously providing antibodies, lab space and facilities to M.C., and Karsten Rippe (German Cancer Research Center Heidelberg) for discussion. We are most grateful to Toyoaki Natsume from the lab of Masato Kanemaki (Center of Frontier Research, National Institute of Genetics, Mishima, Shizuoka Japan) for providing HCT116-RAD21-mAC cells. K.B. was supported by the International Max Planck Research School for Molecular Life Sciences (IMPRS-LS). Microscopic images were acquired at microscopes of the Center for Advanced Light Microscopy (CALM) at the LMU Munich. This work was supported by grants of the Deutsche Forschungsgemeinschaft (GRK1657/TP1C and CA198/9-2) to M.C.C. and by the DFG Priority Program SPP 2202 to H.H. and H.L. K.B. was supported by a grant from the National Human Genome Research Institute (RM1-HG007743-02CEGS - Center for Photogenomics) given to H.L. and H.H. E.L.A. was supported by an NSF Physics Frontiers Center Award (PHY1427654), the Welch Foundation (Q-1866), a USDA Agriculture and Food Research Initiative Grant

(2017-05741), an NIH 4D Nucleome Grant (U01HL130010), and an NIH Encyclopedia of DNA Elements Mapping Center Award (UM1HG009375).

## Author contributions

T.C. and E.L.A. initiated the study; M.C. and T.C. conceived the microscopic experiments together with H.H., K.B., and A.M. K.B., M.C., A.M., and M.G.O. performed experiments shown in Figs. 1–4, 6, 8, and Supplementary Figs. 1a, 2–7. A.M. and K.B. performed live cell and super-resolution/confocal microscopy; H.H. provided input on quantitative image analysis, including statistical analysis; A.M. performed segmentation analyses and V.J.S. 3D image analyses for chromatin density mapping data; M.G.O. performed 3D rendering of nuclei. SM performed RAD21-mClover intensities by high-throughput imaging and DNA Halo experiments with support of M.C.C. shown in Supplementary Figs. 1b and 8. Hi-C data were generated by S.S.P.R. and E.L.A. with experimental support of N.M. (Fig. 5). Repli-Seq data (Fig. 7) were provided by D.M.G. and K.N.K. H.L. provided input for the 3D imaging part and M.C.C. for the replication part. M.C. and T.C. wrote the manuscript with support from all authors, in particular from E.L.A.

## Funding

## Competing interests

The authors declare no competing interests.
