## [Peer Review File · Nature Communications]

Reviewers' comments:

Reviewer #1 (Remarks to the Author):

In the manuscript "Cohesin depleted cells rebuild active and inactive nuclear compartments after mitosis" Cremer et al. explore chromosome architectural changes in HCT116 cells depleted of cohesin. Although this system has been studied in depth by several groups, most of the previous work has been done using Hi-C techniques. The Cremer lab in this case makes use of their well-established microscopy expertise to provide details that were not appreciated before. Among other things they show how A and B compartments are aligned in the absence of cohesin and that sites of active DNA synthesis are enlarged, consistent with the idea that cohesin extrusion generally compacts such domains.

Altogether the paper is straightforward and easy to understand. I do not have any major comments. The data provided will likely help guide future studies on large scale impact brought about by cohesin extrusion during TAD formation.

Reviewer #2 (Remarks to the Author):

Summary

Authors studied the roles of cohesin in 3D chromatin structure using HCT116 RAD21-AID cell line. Live cell imaging of RAD21-depleted cells showed that those cells took much longer time to go through mitosis than control cells and eventually form one multilobulated nucleus. Then they compared the higher order nuclear architecture of the cells among control cells, cells rapidly depleted of RAD21 and cells passed through mitosis in the absence of RAD21. As previously reported, Rad21 depletion resulted in the loss of TAD and loop structure. In contrast, A,B compartment and corresponding histone modification were maintained even in the absence of RAD21. TAD, loop structure and A and B compartment were lost during mitosis and re-established after mitosis. Authors utilised various assay to show that the chromatin organisation including Chromatin territories, Interchromatin compartment, DNA replication pattern and timing can be re-established in the absence of cohesion. Interestingly, replication domain is slightly larger than those of control cells and chromatin is less robust in the cohesion deplete cells.

Their approach to use cohesion-depleted cells passed through mitosis to examine the role of cohesin for the re-establishment of 3D chromatin structure is interesting and their multiple approach to address the issue is impressive. However, there are not more mechanistic insight was addressed in this MS. It has been already known that depletion of cohesion even at G1 phase do not change replication pattern and also do not affect A and B compartments. I also found that the authors are often not providing enough information to support their claims (see below for individual points). Especially live cell imaging of chromosome DNA is not sufficient to claim prolonged telophase and lack of cytokinesis. Therefore I felt these findings are not sufficient or exciting enough for the publication in Nature Cell Communication but could be suitable for more specialised journal.

Points:

In general

In order to prove that the data is reproducible, the authors need to state that how many biological replicate experiments were performed for each figure.

Supple Fig.2

Quantification of RAD21 depletion by flow cytometry (to show cell to cell variation) and western blotting (to show overall level) is required (Supple Fig. 2).

Figure 1

A, B) Please change the colour from red to white. Red on black background is difficult to see.

Authors need to show the data for individual cells (how many hours in mitosis etc).

C) Authors stated that telophase last 30 min (line 174) and cytokinesis did not occur in auxin treated cells. They need to show the evidence how do they categorise telophase/cytokinesis cells, possibly by immuno staining or live cell imaging of the outline of cells.

Figure 2C

Chromosomes in Rad21-depleted cells appeared to too less defective compared to the live cells shown in Figure 1. Please confirm if it is a representative image. It is possible that the depletion was incomplete in the imaged cell (This question is related to my request above to quantify the depletion level as a population and also as individual cell).

Figure 2E

The authors need to show how frequently they see MLN cells with >4painted region and measure the DNA contents in individual MLN cells. This is because nuclei size shown in 2E is almost double of the ones in 2D. How can it be possible? Did they pass mitosis twice? Please check if there are any correlation between the size of nuclei and the number of CTs.

Figure 3

Show the representative IC and N regions in A and B, too as N region in C can't be clearly seen. What is the authors explanation of relative increase of 1 and 2 region in cohesin-depleted cells? It could be simply due to the enlarged nucleus volume in cohesin-depleted cells.

Figure 9

My impression from 9A does not match to the numbers shown B and C. Authors need to include enlarged image and show the number for individual cells and size distribution. How many times the experiment repeated? The result of cells at mid/late S-phase could be useful to confirm their findings.

Discussion

Authors stated in Line 447 that Our Repli-Seq data and live cell observations confirm an undisturbed cell cycle progress of cohesin-depleted cells towards mitosis. However, these data are not sufficient to confirm that cell cycle progression is not perturbed until mitosis entry in auxin treated cells. At least, they need to show the data that how long did the individual control cells and auxin-treated cells take to enter mitosis from the start of live cell imaging. Furthermore, Live cell imaging alone is not sufficient to claim that auxin treated cells arrested of telophase with lack of cytokinesis.

There is no meaningful discussion for MLN formation can be made without confirming above statement of telophase/cytokinesis.

Too much spaces are used to summarise previous studies and the discussion related to MS

Reviewer #3 (Remarks to the Author):

Cremer et al, Cohesin depleted cells rebuild active and inactive nuclear compartments after mitosis

In this manuscript, Cremer et al. used the AID protein degradation system to conditionally deplete the cohesin subunit, RAD21, in human HCT116 cells and explored the changes in chromatin organization in cohesin-depleted cells using live imaging, confocal fluorescent microscopy, 3D-SIM super-resolution microscopy, Hi-C, Repli-seq, and DNA replication labeling. What the authors have claimed to have found can be summarized as follows: (1) cohesin depletion led to delayed mitosis and frequent generation of a single postmitotic multi-lobulated nucleus (MLN), (2) chromosome territories (CTs) were maintained in cohesin-depleted MLNs, (3) proper segregation of active and inactive nuclear compartments was observed in cohesin-depleted MLNs as assayed by nuclear

DAPI intensity distribution, (4) interchromatin compartment was maintained in cohesin-depleted MLNs, (5) proper distribution of different nuclear markers (SC35+ nuclear speckles, transcription initiation form of RNA pol II (Ser5 phosphorylation+), and H3K27me3+) was observed in cohesin-depleted MLNs, (6) Hi-C revealed the elimination of chromatin loops but maintenance of A/B compartmentalization in cohesin-depleted pre- and post-mitotic cells, (7) spatio-temporal patterns of DNA replication was maintained in cohesin-depleted MLNs, (8) Repli-seq revealed the maintenance of early/late replication timing patterns suggestive of the maintenance of A/B compartmentalization in cohesin-depleted pre- and post-mitotic cells, and (9) individual replication foci in early S-phase became decompacted upon cohesin depletion.

Overall the manuscript is descriptive but very comprehensive with regards to the types of observation made in cohesin-depleted cells, utilizing a wide range of cutting-edge technologies listed above. However, detailed descriptions are often missing in the figure legends, which makes it difficult for the readers to interpret the data correctly. In addition, the overall impression of the paper is somewhat different from what the title and the abstract suggests, and this becomes evident especially after reading the Discussion section, which is mostly dedicated to the discussion of different types of 'chromosomal domains.' Disappointingly, the discussion is rather superficial, not necessarily evidence-based, and there is a lack of coherency. I describe my concerns below.

Major points:

1. Mixed use of terms A/B 'compartment domains' and 'compartmental domains'

Even in the postmitotic MLNs generated by cohesin-depletion, global chromatin compartmentalization was maintained, even though loop domains disappeared. Consistent with the maintenance of nuclear compartments, spatio-temporal patterns of replication domains (RDs) or replication foci (RFi) in the nucleus was maintained. What was surprising was the authors' finding that individual RDs or RFis maintained their structure in cohesin-depleted nuclei, although they were slightly expanded in shape (Fig. 9C and Fig. S5). So how do you interpret the identity of these RDs or RFis observed in cohesin-depleted cells? In the Introduction section, the authors claimed that RDs correspond to compartment domains, which are maintained even after cohesin depletion (Fig. 8 and consistent with previous works). However, in the Discussion section, the authors claimed that the boundaries of RDs aligned with the boundaries of compartmental domains, although the authors never provided any experimental data on 'compartmental domains' in the manuscript. This is very confusing and the authors should correct this.

From the viewpoint of the size range and the number of domains, I am afraid the authors are incorrect. The definition of 'compartmental domains' by Victor Corces in his 2018 Nature Review Genetics article is that they are domains identified by PCA of high-resolution Hi-C data with a bin size in the range of a few kb. Thus, they must be much smaller than TADs but their average counts only slightly increased from $\sim 3,600$ to $\sim 5,000$ even 30h after auxin treatment, which seems contradictory if these RDs/RFis in cohesin-depleted cells were really compartmental domains.

RDs or RFis observed in cohesin-depleted nuclei shown in Fig. 9 are not A/B compartment domains either, because compartment domains are usually larger than 1 Mb and sometimes can be up to ~ 10 Mb. You cannot possibly have $\sim 5,000$ compartment domains on the human genome, as shown in Fig. 9B. The authors should discuss the identity of these RDs observed in cohesin-depleted cells more deeply, in a more evidence-based manner, with more careful use of technical terms (compartment domains and compartmental domains) so as not to confuse the readers. What is described on page 23-24 (lines 502-508) is completely beyond me.

2. Use of various domain terms without defining them

This makes it very difficult to read the manuscript. The authors should try to provide some definition and avoid relying entirely on references when a new term first appear in the manuscript. Just to give some examples:

p. 3 \sim , contact domains, loop domains, compartment loops

p. 4 \sim , chromatin domains, chromatin domain clusters, replication domains

p. 23~, metaTADs, subTADs, compartmental domains

3. Lack of coherency

The points I raised above (#1 and #2) are very important for the readers to understand the authors' argument. However, addressing these points would also make the overall content of the paper even more dissociated from the current title and the abstract. I suggest the author to reconsider the title and the content of the abstract for coherency.

4. p.13, Fig. 5A–C: The SC35 signal distribution seems different between the control and the auxin-treated cells. Are they really the same as the authors claimed? Any explanation?

5. p.14, line 322: I did not see any data in Fig. 6 that supports the following statement: "A heightened compartmentalization was noted in particular in B-type chromatin of MLN." The same is true for the description from line 328 to 331 regarding the identification of a histone modification cluster that corresponded to the positions of this particular B-type subcompartment.

6. p.15, Fig. 6A and 6C: I did not understand the meaning of the '-auxin' data, i.e. the left-bottom heatmap in both Fig. 6A and 6C. The same is true for cluster 4 in Fig. 6C and 6D. Also, what do the red square numbers (legend embedded within Fig. 6C) mean? Why does the Hi-C profile of the '+auxin (28h)' cells appear low resolution in Fig. 6C? Please explain.

7. Fig. 6B: the '+auxin (28h)' cells must include a significant number of non-MLN cells (~40%). How can the authors claim that postmitotic MLNs reconstituted the A/B compartments? (p. 14, line 321)

8. Fig. 8C: The claim is unclear: the authors need to explain more about this figure. The same can be said about the main text (p. 18, line 388–393): the description is clearly inadequate. At present, I don't see anything convincing that supports the authors' conclusion that there is strong correspondence between replication timing and genome compartmentalization with A and B compartments before and after cohesin depletion. Also, what are the arrows in Fig. 8C?

9. Fig. 9A and 9B: The number of foci in the '30h auxin' nuclei (Fig. 9A) looks smallest among the three images by visual inspection but Fig. 9B says otherwise. Also, can you zoom in on the individual foci in Fig. 9? I say this because at this resolution, it is difficult to see whether these individual foci really expanded in volume as the bar plot in Fig. 9C suggests.

Minor points:

p. 2, line 45–48: long and wordy sentence

p. 4, line 134: what do you mean by "two structures"?

p. 4, line 139: I didn't understand the following sentence: "However, we find that the physical size of replication foci is smaller." Maybe the authors meant "larger in cohesin-depleted cells?"

p. 5, line 162: life > live

p. 6, line 175: seemingly > seeming

p. 9, Fig.2C and 2D, the authors should show the ratio of abnormal mitosis.

p. 17, Fig. 7D: the legend says 'mid to late' but it only contains 'early' and 'mid' foci patterns. Actually, the figure format makes it very difficult for the readers to delineate Fig. 7C, 7D, and 7E.

p. 19, line 400: replication and timing -> replication timing

p. 21, line 442: What is the conclusion of the DNA halo assay? The authors could add a sentence at the end of the last paragraph of the Results section.

Reviewer #4 (Remarks to the Author):

This manuscript studies the long-term effects of cohesin depletion on human colon cancer derived cell line HCT116RAD21-mAC. Cohesin depletion is achieved by addition of auxin, which in this cell line results in disintegration of cohesin from chromatin. The long term effects are delayed mitosis and multilobulated nucleus (MLN). Various experiments were performed by live-cell spinning disk confocal and fixed cell SIM super-resolution microscopy, as well as Hi-C and Repli-Seq, comparing control cells that underwent normal mitosis with the MLN cells, and find that despite cohesin depletion there are no significant differences in microscopic organisation of nuclear material.

For me it is not immediately clear what is novel in this study and what the information gained in this study is applicable to. Why are the MLNs the focus of this study, since these do not progress through mitosis normally? The title states "Cohesin depleted cells rebuild active and inactive nuclear compartments after mitosis" but why is this important? Would be helpful if this was stated clearly in the introduction. The discussion mentions some situations where MLN occur but leaves it unclear whether these results are relevant to these conditions and, for example, treatment of any disease.

None of the microscopy and analysis methods are new. The authors state that "With super-resolved microscopy we demonstrate that nuclei of pre- and postmitotic cohesin depleted cells maintain principal structural features of the ANC-INC model" - this has been shown to be true for different cell types in refs 14, 15 using the exactly same microscopy and analysis methods as in this paper. While these studies do not look at cohesin depletion, ref 14 states "Notably, these principal structural features of the ANC-INC model are also maintained in cohesin depleted nuclei [93] [94] and even reconstituted in these cells after mitosis [93]". Therefore I am not sure what is the novelty of the current study?

The authors use 3D SIM to measure chromatin compaction of nuclei. This is done by classification of DAPI intensity in the 3D-SIM images in seven classes. This method is described in ref 22 and has been used in other studies e.g. 14, 15. The results in Fig 3 show similar topography of chromatin compaction in control and cohesin depleted nuclei. Would this not be expected?

In Fig 4 the authors aim to demonstrate IC-channels extending between clusters of lamina associated domains into the nuclear interior. These images do not have sufficient resolution to make this conclusion. Figs 4a-c appear to show some vertical channels but it is also possible that these images contain some artefacts as these vertical stripes are very regular and prominent. Why are these channels always vertical, including at the ends of the nucleus where I would expect to see horizontal channels towards the centre? Figs 4d-f show areas from a-c with intensity classification which does show some of these channels (in darker areas of the images) extending to the centre, but in brighter areas these are missing after the intensity classification, further indicating that the intensity variations could be caused by other sources e.g. image reconstruction.

In Fig 5 it is difficult to see where exactly the SC35, H3K27me3 and active RNA Pol II are located. For example, the authors state "RNA Pol II was enriched in the PR, i.e. decondensed chromatin lining the IC" but it is very difficult to assess whether this is true from the images that are shown. I would suggest removing DAPI from the inset images left images (while showing the DAPI classified image next to it) to make this clearer.

In Fig 9 please show enlarged example areas of segmentation.

Point-by-point answers to reviewer's comments

General remarks:

- We are indebted to our four reviewers for their constructive and detailed comments, which prompted a major revision of the manuscript, including the entire text, figures and extended materials. We hope that this revised version fully justifies the major conclusion of the title:

Cohesin depleted cells rebuild functional nuclear compartments after endomitosis

- Compared with the title of the previous submission 'Cohesin depleted cells rebuild active and inactive nuclear compartments after mitosis', we changed 'mitosis' into 'endomitosis' to emphasize that cohesin-depleted cells lack the ability to pass through normal mitosis, including karyokinesis and cytokinesis, but form a single post-mitotic cell with a multi-lobulated nucleus (MLN). The words 'active and inactive' were replaced by 'functional' in line with our demonstration that MLN retain the ability to enter and pass through S-phase with typical early, mid and late-replicating pattern of replication domains (RDs) and a concomitant increase of the DNA content from 2n to 4n.
- In view of the profound differences in architecture between chromosomal territories during interphase and mitotic chromosomes, the ability of endomitotic cells to re-build basic features of the compartmentalized functional nuclear architecture described by the ANC-INC model despite the continuous absence of cohesin, was by no means an obvious, a priori expected result.
- The current study provides the most detailed comparison of quantitative 3D image analyses so far of cohesin-depleted post-endomitotic MLN with nuclei of pre-mitotic cohesin-depleted cells and control nuclei with concurrent Hi-C analyses.
- An integrative presentation and discussion of microscopic and Hi-C results is complicated by the current lack of an overarching nomenclature on chromatin structures on which microscopists, biophysicists, biochemists and molecular biologists would agree. In the new Supplemental Table 1 we define terms used throughout the text for structural and functional entities of higher order chromatin organization obtained by microscopic or Hi-C and related methods. We hope that this additional information is helpful to eliminate confusions of terminology raised by our reviewers.

The revised version contains 8 Figures, 9 S. Figs and 3 S. Tables

revised version	first submission	short caption
Fig. 1 (revised)	Fig. 1	Live cell observations
Fig. 2 (revised)	Fig. 2	chromosome territories (CTs)
Fig. 3 (revised)	Fig. 3	topography of CDCs co-aligned with the IC
Fig. 4 (revised)	Fig. 5	3D topography of functional nuclear markers
Fig. 5 (revised)	Fig. 6	Hi-C data indicating elimination of chromatin loops, but maintenance of A/B compartments after mitosis
Fig. 6 (revised)	Fig. 7	replication patterns
Fig. 7 (revised)	Fig. 8	Hi-C/Repli-Seq demonstrating maintenance of replication timing
Fig. 8 (revised)	Fig. 9	Segmentation of individual replication domains
Supplementary Fig. 1	Supplementary Fig. 1	Scheme of auxin induced RAD21 proteolysis
Supplementary Fig. 2 (revised)	Supplementary Fig. 2	RAD21-mClover proteolysis under auxin treatment
Supplementary Fig. 3 (new),	includes parts from former Suppl. Fig. 2	Time course and quantitative measurement of RAD21 degradation from single cell analyses
Supplementary Fig. 4	Supplementary Fig. 3	mitotic index/ fraction of MLN after cohesin depletion
Supplementary Fig. 5	Supplementary Fig. 4	selective presentation of CT12
Supplementary Fig. 6 (new)	----	3D topography for functional nuclear markers in different series
Supplementary Fig. 7(new)	----	DNA content measurements in controls and cohesin-depleted postmitotic MLN
Supplementary Fig. 8 (revised)	Fig. 4	3D network of an IC channel system
Supplementary Fig. 9 (revised)	Supplementary Fig. 5	DNA halo experiments
Supplementary Table 1 (new)	---	notes on terminology for chromatin structures based on microscopic and biochemical evidence
Supplementary Table 2 (new)	----	mitotic outcome in live cell observations
Supplementary Table 3 (revised)	Supplementary Table 1	statistics

We wrote our answers to reviewer's comments with the intention that they can be read by each reviewer
independently, but we also wanted to avoid extensive repetitions. Accordingly, we refer at appropriate
places to line numbers of this **Point-by-Point Letter** (PbP lines) and to line numbers of the **Revised MS**
(MS lines).

**Reviewer #1:**

*In the manuscript "Cohesin depleted cells rebuild active and inactive nuclear compartments after mitosis"*
*Cremer et al. explore chromosome architectural changes in HCT116 cells depleted of cohesin. Although*
*this system has been studied in depth by several groups, most of the previous work has been done using*
*Hi-C techniques. The Cremer lab in this case makes use of their well-established microscopy expertise to*
*provide details that were not appreciated before. Among other things they show how A and B*
*compartments are aligned in the absence of cohesin and that sites of active DNA synthesis are enlarged,*
*consistent with the idea that cohesin extrusion generally compacts such domains.*
*Altogether the paper is straightforward and easy to understand. I do not have any major comments. The*
*data provided will likely help guide future studies on large scale impact brought about by cohesin extrusion*
*during TAD formation.*

Answer: We thank reviewer #1 for this positive assessment.

**Reviewer #2:**

Summary

*Authors studied the roles of cohesin in 3D chromatin structure using HCT116 RAD21-AID cell line. Live*
*cell imaging of RAD21-depleted cells showed that those cells took much longer time to go through mitosis*
*than control cells and eventually form one multilobulated nucleus. Then they compared the higher order*
*nuclear architecture of the cells among control cells, cells rapidly depleted of RAD21 and cells passed*
*through mitosis in the absence of RAD21. As previously reported, Rad21 depletion resulted in the loss of*
*TAD and loop structure. In contrast, A,B compartment and corresponding histone modification were*
*maintained even in the absence of RAD21. TAD, loop structure and A and B compartment were lost*
*during mitosis and re-established after mitosis. Authors utilised various assay to show that the chromatin*
*organisation including Chromatin territories, Interchromatin compartment, DNA replication pattern and*
*timing can be re-established in the absence of cohesion. Interestingly, replication domain is slightly larger*
*than those of control cells and chromatin is less robust in the cohesion deplete cells. Their approach to*
*use cohesion-depleted cells passed through mitosis to examine the role of cohesin for the re-*
*establishment of 3D chromatin structure is interesting and their multiple approach to address the issue is*
*impressing. However, there are not more mechanistic insight was addressed in this MS. It has been*
*already known that depletion of cohesion even at G1 phase do not change replication pattern and also do*
*not affect A and B compartments.*

A: Previous studies (Oldach et al., 2019; Sima et al., 2019; Rao et al., 2017; Haarhuis et al., 2017) that
already demonstrated the maintenance of A and B compartments and of unchanged spatio-temporal
patterns of replication domains during the ongoing interphase of cohesin depleted cells were cited in our
manuscript. Using a combination of Hi-C analysis, live cell observation and super-resolved microscopy,
we made several so far unreported and unexpected observations: 1) Our prolonged observation periods
revealed that cohesin depleted nuclei are able to pass through (endo)mitosis with the formation of a single
multilobulated nucleus (MLN). 2) Our study led us conclude that cohesin is not decisive for mechanisms,
which instruct the rebuilding of a compartmentalized nuclear architecture after mitosis despite the
continued absence of chromatin loops and TADs. The resulting MLN re-build a compartmentalized nuclear
architecture (chromosome territories, chromatin domain clusters, A/B compartments, replication domains)
with the ability to accomplish another round of S-phase. The detailed nuclear landscape of MLN shows a
full restoration of structural and functional features described by the ANC-INC model established on
microscopic data and A and B compartments based on Hi-C. We argue for concordance between the A
compartment (Hi-C) and the ANC (active nuclear compartment), and between the B compartment and the
INC (inactive nuclear compartment). Quantitative image analyses of nuclear serial sections obtained with
3D SIM demonstrates that ANC/INC compartments, respectively, are constituted by chromatin domain
clusters (CDCs) with a zonal organization of repressed chromatin domains in the interior and
transcriptionally competent domains located at the periphery that are lined by a channel system, called the
interchromatin compartment (IC) with its own functions as storage and transport system. (3) In
experiments, where we compared numbers and volumes of segmented replication domains between
control nuclei grown for 6h after replication labeling, pre-mitotic nuclei grown for 6h in auxin after RL and

post-endomitotic MLN grown for 30h in auxin after RL, we noted an overall increased average RD volume
and a higher heterogeneity of RD volumes. These results argue that cohesin is required for the full
structural maintenance at the level of RDs. Although we agree with reviewer #2 that our manuscript does
not provide additional mechanistic insights on cohesin, whose functions are still incompletely understood,
we hope, as pointed out by reviewer #1, that our data should “help guide future studies on large scale
impact brought about by cohesin extrusion during TAD formation.”

*I also found that the authors are often not providing enough information to support their claims (see below*
*for individual points). Especially live cell imaging of chromosome DNA is not sufficient to claim prolonged*
*telophase and lack of cytokinesis. Therefore I felt these findings are not sufficient or exciting enough for*
*the publication in Nature Cell Communication but could be suitable for more specialised journal.*

*Points:*

*In general*

*In order to prove that the data is reproducible, the authors need to state that how many biological replicate*
*experiments were performed for each figure.*

*A: All microscopic observations were verified from at least two independent series with exception of Fig. 8*
*(see below). For highly elaborate quantitative 3D analyses of super-resolved image stacks with each*
*nucleus comprising between ~70-100 serial sections we selected between 4 and 22 nuclei from a given*
*experimental series with the only precondition of a high staining and structure-preserving quality. Data*
*shown in Figs. 3 and 4, Supplementary Figs. 2 and 9 comprise merged data from different series, with*
*links to data on individual experiments. A link for informations on additional experiments and „biological*
*replicate experiments“ and their access is provided in the respective Figure legends and also indicated in*
*the section „data availability“ in Material &Methods; (MS lines 924ff). For Hi-C related experiments shown*
*in the revised Figs. 5 and 7 (at least) 2 replicates of each timepoint were performed. This information was*
*provided in the nr-reporting summary of the previously submitted version and is now also provided in*
*Material &Methods in section „data availability“.*

*Supple Fig.2*

*Quantification of RAD21 depletion by flow cytometry (to show cell to cell variation) and western blotting (to*
*show overall level) is required (Supple Fig. 2).*

*A: A meticulous check for effectiveness of the auxin induced degron (AID) system used in our study is*
*pivotal and the suggested quantification of RAD21 depletion by western blotting (global level) and/or flow*
*cytometry (single cell level) could provide relevant information. Due to restrictions and lockdown related to*
*the COVID-19 pandemic, we lacked access to a flow cytometry facility. As an alternative, we took*
*advantage from our live cell time lapse data shown in Fig. 1 and Supplementary Fig. 3 to perform*
*quantitative measurements of RAD21 decay on a single cell basis. In addition, we provide high throughput*
*data with measurements of RAD21-mClover intensities in fixed cells, both of controls and after cohesin*
*depletion. We consider these analyses adequate to the reviewer’s suggestion. These quantitative data of*
*RAD21 degradation are now added in Supplementary Fig.2.*

*Please note that we checked loss of RAD21-mClover fluorescence of each auxin treated nucleus included*
*in an evaluation except for 3D FISH experiments (MS lines 135ff).*

*Figure 1*

*A, B) Please change the colour from red to white. Red on black background is difficult to see. Authors*
*need to show the data for individual cells (how many hours in mitosis etc).*

*A: We agree that b/w representation provides clearer delineation of structures and have changed the color*
*scheme. We also added detailed data of all individual recorded control and cohesin depleted cells that*
*were followed through mitosis (Supplementary Table 2).*

*C) Authors stated that telophase last 30 min (line 174) and cytokinesis did not occur in auxin treated cells.*
*They need to show the evidence how do they categorise telophase/cytokinesis cells, possibly by*
*immunostaining or live cell imaging of the outline of cells.*

*A: We agree that a precise assignment of mitotic stages by visual inspection is questionable without*
*additional delineation of mitosis-associated markers (such as tubulin, kinetochores, aurora B or other parts*
*of the chromosomal passenger complex, compare e.g. Ruchaud et al. (2007) Nature Rev Mol Cell Bio,*
*DOI: 10.1038/nrm2257). Detailed analyses of the course of mitotic progression were however beyond the*

goals of our study which was focused on the potential of cohesin depleted cells for a reconstitution of a
functional nuclear architecture after mitosis. Yet, we clearly demonstrate that chromatid separation occurs
in mitoses of cohesin depleted cells, but despite chromatid separation we never observed subsequent cell
division (cytokinesis) (Fig. 2, Supplementary Table 2, **MS lines 155ff**). Instead, we consistently observed
the outcome of a single postmitotic multilobulated nucleus (MLN) typical for endomitosis. Accordingly, in
Fig 2 (see below) we demonstrate that most MLN harbor four territories for a given chromosome. The
previously described role of cohesin on spindle pole integrity and kinetochore-microtubule attachment is
likely relevant for this pathological outcome. We have revised the text and the legend to 1C accordingly
(**MS lines 157ff**).

*Figure 2C*

*Chromosomes in Rad21-depleted cells appeared to too less defective compared to the live cells shown in*
*Figure 1. Please confirm if it is a representative image. It is possible that the depletion was incomplete in*
*the imaged cell (This question is related to my request above to quantify the depletion level as a*
*population and also as individual cell).*

160 A: For the different appearance between mitotic chromosomes shown in Fig. 1 and in Fig. 2C please note
that the latter are z-projections of complete confocal image stacks from fixed specimen, while mitoses
shown in Fig. 1 were recorded from live cells by spinning disk microscopy. Resolution between a spinning
disk microscope and a point scanning confocal microscope is in principle comparable if -as in our case-
objective magnification matches pinhole size. Long term observations (21h) with a time resolution of 15
165 min, however, require reduction of the excitation light dose to minimize phototoxicity. This results in a
166 reduced signal to noise ratio affecting the outline of mitotic shapes and only allows to acquire one plane
167 per metaphase. As to the question of a possibly incomplete cohesin depletion in metaphases shown in
Fig. 2C: In fact (**stated in MS lines 135ff**), in 3D-FISH experiments auxin-induced loss of cohesin could not
be verified for individual nuclei since heat denaturation -required for 3D-FISH- degrades the reporter
fluorescence and thus prevents such a verification. Accordingly, auxin-induced cohesin depletion was not
unequivocally proven in metaphases shown in Fig. 2C. Yet, it is very likely that the shown mitoses
including the 'normal looking' in the left image, emerged from cohesin depleted nuclei: (1) Based on our
observation of ~4% "escapers" for RAD21 degradation the random probability that mitoses in Fig. 2C
belong to the small group of AID "escapers" is very small; (2) In our live cell observations most early
mitotic figures emerging from cohesin depleted cells appeared un conspicuous, (compare live cell series
found at <https://cloud.bio.lmu.de/index.php/s/rZxxkgYExonWLgy?path=%2FFig1>), in agreement with
numerous un conspicuous mitotic figures seen in overviews recorded 6h after cohesin depletion
(Supplementary Fig. 4). Notably, formation of un conspicuous bipolar metaphase plates in the beginning of
mitosis of cohesin depleted cells was also reported by *Diaz-Martinez (2010)*. The mitosis in the mid image
of Fig. 2 shows chromatid separation (with a slight lacking behind of single chromatids), while the mitosis
on the right has clearly an abnormal configuration in line with live cell observations where such abnormal
configurations accumulate during the prolonged course towards endomitosis.

*Figure 2E*

*The authors need to show how frequently they see MLN cells with 4 painted region and measure the DNA*
*contents in individual MLN cells. This is because nuclei size shown in 2E is almost double of the ones in*
*2D. How can it be possible? Did they pass mitosis twice? Please check if there are any correlation*
*between the size of nuclei and the number of CTs.*

189 A: Most MLN show 4 CTs for a given painted chromosome. However, in contrast to control nuclei, a
190 fraction of respective CTs (in particular CT19) in MLN show a torn appearance of painted regions
sometimes connected by thin chromatin bridges (compare Supplementary Fig. 5). This complicates an
unambiguous assessment of CT numbers in these nuclei. Possible explanations for this feature of CTs in
MLN such as mechanic forces during lobe formation, a higher level of relaxation/decondensation and
increased mechanical instability of chromosomes in cohesin depleted nuclei are outlined in **MS lines 200**.
Re-inspection of painted MLN did not reveal a correlation between nuclear size and apparent number of
painted regions (or their level of 'tearing-up'). As to the different size of nuclei: In the new Supplementary
Fig. 7 we added a graph with DNA content assessment of MLN compared to control nuclei by
measurements of integrated DAPI intensities. In agreement with the increased size of MLN, we found a
distinctly higher average DNA content in MLN compared to controls. The size increase can at least in part
be explained by our observation that cohesin depleted MLN can pass through another round of replication
(shown in Fig. 6E). This means that post-endomitotic MLN that start after endomitosis with a 2n DNA

content increase their DNA content up to 4n after replication. We did not observe a second mitosis in
MLN. We have changed the text in the results part accordingly (MS lines 196ff) and have also rearranged
Fig. 2 for a better representation of our overall findings on CT configurations and nuclear morphologies.

**Figure 3**
*Show the representative IC and N regions in A and B, too as N region in C can't be clearly seen.*
208 A: We marked regions representative for IC and nucleoli (N) in the revised Fig. 3A and B. Typical nucleoli
in DAPI stained nuclear sections are roundish and surrounded by a rim of compact nucleoli associated
domains (NADs), whereas a typical IC-lacuna shows an irregular rim generated by the perichromatin
region (PR).

*What is the authors explanation of relative increase of 1 and 2 region in cohesin-depleted cells? It could*
*be simply due to the enlarged nucleus volume in cohesin-depleted cells.*

215 A: To verify this statement and to answer this question properly, we decided to repeat the entire
experiment and to increase the number of evaluated nuclei to n= 38/39/33 (versus 13/13/11 in the
previous version). The slight relative increase of DAPI intensity classes 1 and 2 in cohesin-depleted nuclei
noted in experiment 1 with regard to control nuclei was not confirmed in averaged data of experiment 1
and 2 (revised Fig. 3D); also the slight increase of the average nuclear volume measured for cohesin-
depleted nuclei after 6 h auxin treatment compared to control nuclei shown in the first experiment was not
reproduced (revised Fig. 3E). The doubling of the volume of MLN studied after 30 h auxin treatment was
however fully confirmed. This volume doubling corresponds with the increase of the DNA content of these
post-endomitotic MLN compared with both control and pre-mitotic cohesin depleted nuclei (see
Supplementary Fig. 7 for quantitative DNA measurements). In Fig. 3A-C we now included enlargements of
boxed areas, which exemplify the similar appearance of chromatin domain clusters (CDCs) with a zonal
organization of less compact chromatin domains at the periphery (PR, purple and red) adjacent to the IC
(blue) and higher compacted chromatin (yellow and white) located in the CDC interior both in controls,
cohesin depleted nuclei and MLN.

Great care was taken to keep technical parameters for image recording and quantitative image analysis
constant in both experiments. Therefore they are an unlikely source to explain interexperimental
differences. Differences between replicates 1 and 2 representing snap-shots from the respective
experiments with unsynchronized cell cultures may be attributed to unperceived differences of cell culture
conditions. Notwithstanding these differences, both replicates support our major conclusion: Principal
features of a compartmentalized organization with CTs and CDCs, pervaded by the IC in control nuclei
were maintained in pre-mitotic cohesin depleted nuclei and rebuilt in post-endomitotic MLN. The
accordingly revised text is found in MS lines 309ff and 341ff, see also notes addressing interexperimental
variability referring to Fig. 4 in Supplemental Fig. 6.

**Figure 9**
*My impression from 9A does not match to the numbers shown B and C. Authors need to include enlarged*
*image and show the number for individual cells and size distribution. How many times the experiment*
*repeated? The result of cells at mid/late S-phase could be useful to confirm their findings.*

243 A: We added inset magnifications from the SIM sections of the three nuclei (now Figure 8A). This
comment and further comments of reviewer #2 (see PbP below lines 393ff), and of reviewers #3 and #4
(see PbP below lines 590ff,774) prompted us to perform a thorough additional analysis with data shown in
Fig. 8B-C: in Fig. 8B we present the number of RDs counted in each studied nucleus separately (instead
of summarized RD counts for each series of nuclei in the old Figure). In Fig. 8C we provide volume
measurements for individual RDs (instead of the mean total volume of RDs/nucleus in each series). RD
counts and RD volumes were evaluated in nuclei of three cultures: The 'control' culture was fixed 6h after
RL together with the '6h auxin' culture, which was incubated with auxin immediately after RL. The '30h
auxin' culture was fixed after 30h with auxin, when most cells had passed an endomitosis yielding a
multilobulated cell nucleus. It is important to note that an RD pattern generated by pulse labeling in a
given nucleus is maintained after S-phase and after mitosis, independent of the time of fixation during the
post-endomitotic interphase of MLN.

Nuclei from each culture represent randomly chosen samples with typical early replication patterns. We
consider this strategy to identify a sample of nuclei in the three cultures (split from the same mother
culture and kept under strictly the same culture conditions except for the presence or absence of auxin
after RL) appropriate for a quantitative comparison of RD numbers and volumes. Segmentation and

quantification of RDs were based on complete 3D SIM image stacks from each nucleus using a program
provided by Velocity software for object counting and volume measurements (MS lines 868ff). Nuclei with
mid/late S-phase patterns were not included in our 3D analyses since transient stages between mid and
late S-phase make a demarcation of these phases difficult for comparative evaluations. Moreover,
comparative evaluations might become even more complex due to aggregations of RDs into larger units in
late S-phase. Individual RDs labeled during early S-phase are best distinguishable from each other.

The following results were obtained by this extended quantitative analysis on individual RDs (Fig.
8B-C; supplementary Table 3): (1) An increased heterogeneity of RD counts and volumes in cohesin-
depleted nuclei compared with controls. The broader range of volumes may be related to the cell-to-cell
shift of boundaries described for TAD-like domains in cohesin-depleted cells (Bintu et al. 2018). (2) An
increased number of RD counts after cohesin depletion with significantly higher values in MLN (30 h
auxin) compared to control nuclei. An increase of RD counts in MLN was expected: MLN are generated as
a result of an endomitosis with a full separation of sister chromatids where RDs were pulse-labeled in the
previous cell cycle by incorporation of labeled nucleotides into both newly synthesized DNA strands.
These cells entered anaphase, but did not proceed through karyokinesis and cytokinesis. For the (less
pronounced) increase of RD numbers already observed in pre-mitotic cohesin depleted interphase nuclei
(6h auxin) compared to controls the following should be considered: Controls and 6h treated cells were
replication labeled in early S, but likely advanced to late S/G2 after 6h. In late S/G2 cells the two sister
chromatids present in each CT are kept together by cohesin at certain sites, but are untethered at other
sites (Stanyte et al. 2018. J Cell Biol 217: 1985; Takahashi and Hirota. 2018. J Cell Biol 217: 1887; see
Fig. 1 from Takanashi and Hirota added below). An increase of discernible RDs may result from enhanced
untethering of sister chromatids after cohesin depletion. (3) An increased volume of individual RDs.
Statistical comparisons of RD volume measurements were based on large numbers of segmented,
individual RDs (39.334 (control), 31.467 (6h auxin) und 55.153 (30h auxin) and revealed a highly
significant increase of mean RD volumes in cohesin depleted nuclei. This finding was not expected since
a separation of RDs due to sister chromatid untethering should rather result in a decrease of individual RD
volumes. Our observation of a volume increase of RDs supports a role of cohesin in the compaction of
chromatin structures by chromatin loop extrusion (Kim et al 2019) with effect on contact frequencies and
an impact of TAD patterns in ensemble Hi-C experiments. Due to the resolution limit of 3D-SIM (~120 nm
lateral / 300 nm axial) we considered these results as preliminary and did not repeat the experiment: a
fraction of RDs with sizes below this limit would show a putative size reflecting the diffraction limit,
resulting in an overestimate of their volumes. To overcome these method-inherent limitations, imaging
approaches with higher resolution, such as STORM/SMLM or STED are required to further clarify the
influence of cohesin on RD structure. Yet, our findings let us tentatively conclude that cohesin is essential
for the maintenance of a normal compaction state of RDs which we equate with chromatin domains (CDs;
compare Supplementary Table 1) in line with the necessity of cohesin to preserve the TAD pattern
observed in ensemble Hi-C. In summary, we conclude that cohesin plays an indispensable role for the
structure of RDs/CDs, but is dispensable for the formation of a compartmentalized nuclear organization.
These issues are dealt with in MS lines 462-505 and 587-617.

Figure 1. **Dynamic organization of sister chromatids after replication.** Replicated chromatin fibers are highly mobile and readily dissociate from each other for >300 nm, beyond the achievable resolution by light microscopy. Cohesin, which mediates interchromatid (a) and intrachromatid (b) tethering, possibly confers a correct size for sister chromatids in S and G2 phase, forming a prospective dynamic structure that relates to organizing compacted chromosomes in subsequent mitosis.

**Discussion**
*Authors stated in Line 447 that Our Repli-Seq data and live cell observations confirm an undisturbed cell*
*cycle progress of cohesion-depleted cells towards mitosis. However, these data are not sufficient to*
*confirm that cell cycle progression is not perturbed until mitosis entry in auxin treated cells. At least, they*
*need to show the data that how long did the individual control cells and auxin-treated cells take to enter*
*mitosis from the start of live cell imaging. Furthermore, Live cell imaging alone is not sufficient to claim that*
*auxin treated cells arrested of telophase with lack of cytokinesis. There is no meaningful discussion for*
*MLN formation can be made without confirming above statement of telophase/cytokinesis.*

309 A: Our statement of a seemingly undisturbed cell cycle progression of cohesin-depleted cells towards
mitosis refers to a previously published study from *Oldach et al. (2019)* performed on synchronized cells.
We have clarified this in the revised text (MS lines 444, 581ff). The correspondence of the untreated and
auxin-treated Repli-Seq tracks in our Repli-Seq experiments supports this conclusion. It is strengthened
by our live cell observations, which strongly argue against major delays compared with controls. The
drastic accumulation of mitotic cells noted after 6 h auxin treatment (Supplemental Fig. 4) indicates that
cohesin-depleted cells “go easily into mitosis but not easily out”. The combination of two entirely different
approaches, Repli-Seq and microscopy, makes a strong point for undisturbed cell cycle progression of
cohesin depleted cells towards mitosis. With regard to ‘telophase’ arrest, we have modified the text (MS
lines 525ff).

*Too much spaces are used to summarise previous studies and the discussion related to MS.*

321 A: In our experience there is still a substantial lack of integration between microscopic and Hi-C studies.
This gap is exemplified by the lack of a common terminology to describe essential findings of higher order
chromatin organization (see Supplemental Table 1). We revised the entire manuscript with the intention to
introduce, describe and discuss both. Microscopic and Hi-C findings with due consideration of the relevant
literature in ways that are comprehensible for a broad readership.

**Reviewer #3 (Remarks to the Author):**

*In this manuscript, Cremer et al. used the AID protein degradation system to conditionally deplete the*
*cohesin subunit, RAD21, in human HCT116 cells and explored the changes in chromatin organization in*
*cohesin-depleted cells using live imaging, confocal fluorescent microscopy, 3D-SIM super-resolution*
*microscopy, Hi-C, Repli-seq, and DNA replication labeling. What the authors have claimed to have found*
*can be summarized as follows: (1) cohesin depletion led to delayed mitosis and frequent generation of a*
*single postmitotic multi-lobulated nucleus (MLN), (2) chromosome territories (CTs) were maintained in*
*cohesin-depleted MLNs, (3) proper segregation of active and inactive nuclear compartments was*
*observed in cohesin-depleted MLNs as assayed by nuclear DAPI intensity distribution, (4) interchromatin*
*compartment was maintained in cohesin-depleted MLNs, (5) proper distribution of different nuclear*
*markers (SC35+ nuclear speckles, transcription initiation form of RNA pol II (Ser5 phosphorylation+), and*
*H3K27me3+) was observed in cohesin-depleted MLNs, (6) Hi-C revealed the elimination of chromatin*
*loops but maintenance of A/B compartmentalization in cohesin-depleted pre- and post-mitotic cells, (7)*
*spatio-temporal patterns of DNA replication was maintained in cohesin-depleted MLNs, (8) Repli-seq*
*revealed the maintenance of early/late replication timing patterns suggestive of the maintenance of A/B*
*compartmentalization in cohesin-depleted pre- and post-mitotic cells, and (9) individual replication foci in*
*early S-phase became decompacted upon cohesin depletion.*

*Overall the manuscript is descriptive but very comprehensive with regards to the types of observation*
*made in cohesin-depleted cells, utilizing a wide range of cutting-edge technologies listed above. However,*
*detailed descriptions are often missing in the figure legends, which makes it difficult for the readers to*
*interpret the data correctly. The overall impression of the paper is somewhat different from what the title*
*and the abstract suggests, and this becomes evident especially after reading the Discussion section,*
*which is mostly dedicated to the discussion of different types of 'chromosomal domains.' Disappointingly,*
*the discussion is rather superficial, not necessarily evidence-based, and there is a lack of coherency.*

351 A: We appreciate the general assessment of reviewer #3 and his/her further detailed objections. As
indicated on page 1, we revised the entire manuscript, figures and legends. The present study provides an
integrative view on the effects of cohesin depletion on nuclear architecture studied with top-down and
bottom-up approaches. The former approach is an attempt to explore the nuclear landscape as a whole
down to the structural and functional interactions of individual macromolecules, whereas the latter takes
the opposite way to build up a systematic understanding of the integrated structural and functional
organization of the nucleus starting at the molecular level. There is currently a lack of an overarching
nomenclature to describe structural and functional higher order chromatin entities discovered by either
approach. An agreement on a consistent terminology is a necessary step towards a common theoretical
framework. For a definition of terms as we use them throughout the revised manuscript, we refer readers
to the new Supplemental Table 1, see below. We argue that co-aligned active and inactive compartments
(ANC and INC), identified in microscopic studies, correspond with A and B compartments, detected with
ensemble Hi-C. We further suggest to equate replication domains (RDs) and chromatin domains (CDs)
with similarly sized compartment domains (see below) rather than with TADs and hope that our
interpretation is helpful to further the field of 4D nucleome research (MS lines 551ff).

1. Mixed use of terms A/B 'compartment domains' and 'compartmental domains'
Even in the postmitotic MLNs generated by cohesin-depletion, global chromatin compartmentalization was
maintained, even though loop domains disappeared. Consistent with the maintenance of nuclear
compartments, spatio-temporal patterns of replication domains (RDs) or replication foci (RFi) in the
nucleus was maintained.

372 A: We fully agree with the reviewer and apologize for our mixed use of terms A/B compartment domains
and compartmental domains. As reviewer #3 emphasizes, the term 'compartmental domains' refers to
high-resolution Hi-C data with a bin size of ~10 kb in *Drosophila* (Rowley *et al.* 2017; Rowley and Corces
2018). Rowley *et al.*, 2017 equate compartmental domains with A/B compartments at high resolution. In
the revised manuscript we use the term compartment domains to refer to domains which typically
comprise hundreds of kb and are functionally defined by their enrichment with epigenetic marks for
transcriptionally competent (A domains) or repressive chromatin (B domains) which in their entirety form
the A and B compartment. Reviewer #3 prompted us to carefully check the whole terminology currently
used in microscopic and Hi-C studies and to implement this issue in a new Table (Supplementary Table 1
with added references): *Overview and explanatory notes on terminology for higher order chromatin*
*structures*. In this Table we describe the current use of terminologies together with a provisional attempt to
point out relationships between specific terms used in microscopic and Hi-C studies, respectively.

Exact relationships between TADs, CDs, RDs, compartment have remained elusive and these entities
should not be considered as structurally and functionally uniform entities. A compartmental domain
comprising an active and an inactive gene, for example, may be located within a repressed CD providing
an anomaly like a corn of pepper in a sugar box. More detailed comparisons between the nuclear
landscapes present in different cell types and species are required to close this important gap of
knowledge. Structure-function relationships between compartment domains and compartmental domains
could be further resolved with optical reconstruction of chromatin architecture (ORCA), a method that can
trace the DNA path in single cells with nanoscale accuracy and genomic resolution reaching two
kilobases. ORCA would allow to test, whether nucleosome clutches form kb-sized compartmental domains
which in turn form chromatin domains, which we suggest to equate with compartment domains but such
experiments are clearly beyond the scope of our present study (MS lines 618ff).

*What was surprising was the authors' finding that individual RDs or RFis maintained their structure in*
*cohesin-depleted nuclei, although they were slightly expanded in shape (Fig. 9C and Fig. S5). So how do*
*you interpret the identity of these RDs or RFis observed in cohesin-depleted cells? In the Introduction*
*section, the authors claimed that RDs correspond to compartment domains, which are maintained even*
*after cohesin depletion (Fig. 8 and consistent with previous works). However, in the Discussion section,*
*the authors claimed that the boundaries of RDs aligned with the boundaries of compartmental domains,*
*although the authors never provided any experimental Data on 'compartmental domains' in the*
*manuscript. This is very confusing and the authors should correct this. From the viewpoint of the size*
*range and the number of domains, I am afraid the authors are incorrect. The definition of 'compartmental*
*domains' by Victor Corces in his 2018 Nature Review Genetics article is that they are domains identified*
*by PCA of high-resolution Hi-C data with a bin size in the range of a few kb. Thus, they must be much*
*smaller than TADs but their average counts only slightly increased from ~3,600 to ~5,000 even 30h after*
*auxin treatment, which seems contradictory if these RDs/RF in cohesin-depleted cells were really*
*compartmental domains. RDs or RFis observed in cohesin-depleted nuclei shown in Fig. 9 are not A/B*
*compartment domains either, because compartment domains are usually larger than 1 Mb and sometimes*
*can be up to ~10 Mb. You cannot possibly have ~5,000 compartment domains on the human genome, as*
*shown in Fig. 9B. The authors should discuss the identity of these RDs observed in cohesin-depleted cells*
*more deeply, in a more evidence-based manner, with more careful use of technical terms (compartment*
*domains and compartmental domains) so as not to confuse the readers.*

415 A: In the revised manuscript we have resolved the confusion caused by our mixed use of compartment
domains and compartmental domains. In line with our reasoning that RDs, CDs and compartment
domains have a DNA content of hundreds of kb (see Supplemental Table 1), our average counts of
segmented RDs in individual control nuclei, cohesin depleted nuclei after 6 h auxin treatment, and MLN
after 30 h auxin treatment appear reasonable (Fig. 8). These counts agree well with estimates for both
TADs and compartment domains comprising several hundred kilobases up to ~1 Mb (MS lines 563ff). A
correspondence of microscopically discernible RDs with TADs mapped by ensemble Hi-C has been
favored in some publications (Moindrot *et al.* 2012; Pope *et al.* 2014). It is of note that the definition of

~1Mb chromatin domains (CDs) in our early microscopic studies was based on replication domains (RDs).
RDs were first detected in mammalian cells by pulse-labelling of replicating DNA during S-phase with
halogenated nucleotides. RDs were optically resolved down to a few single replicons (150–200 kb)
clustered per replication site by super-resolution microscopy (Xiang et al. 2018).

In the experiment shown in the revised Fig. 8, replication labeling was performed prior to auxin
treatment. In case of control nuclei, pulse-labeled in early S-phase, each active RD contains pulse-labeled
DNA from both daughter strands. RDs are stably maintained over subsequent cell cycles independent of
their association with the transcription machinery. As a result, an RD pattern labeled in early S-phase, can
be detected after mitosis at all stages of the subsequent interphase. The mechanisms which form and
maintain RDs together as structural and functional units are not clear. In daughter nuclei of control cells,
we would expect a similar number of RDs. Their volume, however, should be smaller than in pre-mitotic
nuclei. Some spatial separation of sister chromatids may already occur in CTs during G2 which may
explain the range of RD counts. For an interpretation of the increase of counts from ~3,600 in control
nuclei to ~5,000 in MLN studied 30h after auxin treatment (Fig. 8A) we consider the following possibilities:
First, an increase of RD counts may be enhanced by an increased untethering of labeled sister chromatids
in pre-mitotic cohesin depleted cells and even more by the spatial separation of sister chromatids in post-
endomitotic MLN. Second, this increase may suggest some disintegration of RDs into smaller subunits
taking place in the absence of cohesin. In this context, the cell to cell shift of boundaries described for
TAD-like domains in cohesin depleted cells may contribute to an increased heterogeneity of RDs in
cohesin depleted nuclei compared with controls.

Our observation of a volume increase (Fig. 8B) was unexpected. The concordant increase of RD counts
and overall volume distributions of individual segmented RDs in MLN cannot be fully explained at present
and should be interpreted with caution. Due to the resolution limit of 3D-SIM (~120 nm lateral / 300 nm
axial), a fraction of RDs with sizes below this limit would show a putative size reflecting the diffraction limit,
resulting in an overestimate of their volumes (as indicated by the fact that the smallest volumes
determined for segmented RDs were the same in controls and cohesin-depleted nuclei). To overcome
these method-inherent limitations, imaging approaches with higher resolution, such as STORM/SMLM or
STED are required (for revised text see MS lines 462ff and 587ff).

*What is described on page 23-24 (lines 502-508) is completely beyond me.*

*A: We deleted this paragraph in line with a revision of former Fig. 8, now revised Fig. 7.*

Panels A and B in this revised Figure support the conclusion that global replication timing noted in controls
(without auxin) is maintained in auxin-treated cells as shown in (A) by the correspondence of the untreated
and auxin-treated Repli-Seq tracks for chromosome 8 together with contact matrices of chromosome 8 at
500 kb resolution and in (B) by the scatter plot of replication timing (percentile of E/L ratio). The new panel
C supports the tight relationship between genome A/B compartmentalization and replication timing, which
is maintained in the absence of cohesin. The top part presents the overlay of nearly identical tracks of
replication timing data for human chromosome 3 ($\log_2(E/L)RT$) from cells cultured with auxin for 6 h (blue)
and control cells without auxin (black). The bottom part shows the result of a principle component 1
analysis of Hi-C data for chromosome 3 (PC1, also known as Eigenvector). Data were based on at least
two replicates of each timepoint.

*2. Use of various domain terms without defining them*

*This makes it very difficult to read the manuscript. The authors should try to provide some definition and*
*avoid relying entirely on references when a new term first appear in the manuscript. Just to give some*
*examples:*

*p. 3~, contact domains, loop domains, compartment loops*

*p. 4~, chromatin domains, chromatin domain clusters, replication domains*

*p. 23~, metaTADs, subTADs, compartmental domains*

*A: We agree with these concerns, which prompted us to compile Supplemental Table 1 and revised the*
*whole manuscript accordingly. We refer to terminology issue discussed above (PbP lines 21ff, 321ff, 359ff,*
*414ff). We hope that this Table facilitates the readability of the manuscript without a disproportionate*
*lengthening. All abbreviations are listed on top of the manuscript.*

*3. Lack of coherency*

*The points I raised above (#1 and #2) are very important for the readers to understand the authors'*
*argument. However, addressing these points would also make the overall content of the paper even more*

*dissociated from the current title and the abstract. I suggest the author to reconsider the title and the*
*content of the abstract for coherency.*

482 A. After careful reconsideration, we trust that the revised title “Cohesin depleted cells rebuild functional
nuclear compartments after endomitosis” emphasizes our major conclusion: based on our major revision
of the entire manuscript (see detailed answers above). Our study demonstrates that cohesin is not
essential for mechanisms, which instruct the rebuilding of a compartmentalized nuclear architecture in
cohesin-depleted cells after passing through an endomitosis, yielding a single daughter cell with a
multilobulated nucleus (MLN).” We replaced ‘mitosis’ by ‘endomitosis’ to be more precise and also
replaced active and inactive compartments with functional compartments. In light of the fundamental roles
ascribed to cohesin, the capacity of MLN to initiate another round of DNA replication with stage-specific
patterns of replication domains (RDs) was not expected. The abstract was revised for coherency.

*4. p.13, Fig. 5A–C: The SC35 signal distribution seems different between the control and the auxin-treated*
*cells. Are they really the same as the authors claimed? Any explanation?*

495 A. This comment prompted a re-examination of 3D SIM images of DAPI stained nuclei with immuno-
496 detection of SC35 and the decision to repeat the entire experiment and thus to increase the number of
497 evaluated nuclei up to n= 38/39/33 (versus 13/13/11 in the previous version). These comprehensive data
are now shown in Fig. 4, separate data for each experiment are provided in supplementary Fig. 6 with the
following conclusions: By a careful re-examination of images we saw that splicing speckles identified by
SC35 appear in somewhat different conformations, either as very compact structures or with a rather
loose dot like appearance, both in nuclei of control cells, as well as in cohesin-depleted pre-mitotic nuclei
and post-endomitotic MLN. For better comparability we have replaced the image in panel B and C of Fig.
4, and in addition provide examples of different SC35 ‘conformations’ in Supplementary Fig. 6. Different
structural conformations of SC35 as integral part of splicing speckles were described (Fei et al. JCS 2017)
but underlying reasons have remained elusive, in particular a correlation with cell cycle was not found. In
the first experimental series most speckles in cohesin depleted nuclei showed a compact conformation
apparently with an almost exclusive localization within class 1 (interchromatin compartment) in contrast to
control nuclei. After evaluating a larger number of nuclei merged from two different experiments (Fig. 4)
including ‘compact’ and ‘loose’ SC35 conformations both in controls and cohesin depleted nuclei (shown
in supplementary Fig. 6) the significant difference between controls and cohesin depleted cells with regard
to the structure of splicing speckles and their relative distribution on DAPI intensity classes was not
confirmed. Since great care was taken to keep technical parameters for image recording and quantitative
image analysis constant in both experiments, we suggest that the interexperimental differences of the
results presented in Supplemental Fig. 6 reflect unperceived biological differences between cultures
studied in the two experiments, which were performed ~12 months apart in the same lab. Although culture
conditions were the same, it is well known that the physiology of cultured cells can be influenced by
various parameters. The revised text is found in **MS lines 291-350**.

*5. p.14, line 322: I did not see any data in Fig. 6 that supports the following statement: "A heightened*
*compartmentalization was noted in particular in B-type chromatin of MLN." The same is true for the*
*description from line 328 to 331 regarding the identification of a histone modification cluster that*
*corresponded to the positions of this particular B-type subcompartment.*

We thank the reviewer for this helpful comment. We have now added the eigenvector of the Hi-C matrix
displayed in Figure 5C (previously Figure 6C) in order to clarify the relationship between the
subcompartment that we annotated on the basis of k-means clustering of histone modification data and
A/B type compartmentalization. As can be seen by comparing the eigenvector and the cluster annotation
track above the contact matrices in 5C, the loci that demonstrate significantly strengthened compartment
interactions line up with both one of our annotated histone modification clusters (cluster 4, annotated in
yellow) and negative values in the eigenvector (i.e. B-compartment) indicating that these loci correspond
to a B-type subcompartment. We have also added figure 5D that illustrates the ratio of average between-
cluster contact probabilities post- and pre- cohesin degradation genome-wide. This systematic analysis
demonstrates both the heightened compartmentalization present in MLN (after 28hrs of cohesin
degradation) vs untreated cells as evidenced by the enrichment of within-cluster contacts after cohesin
degradation. Additionally, this figure highlights the particular enrichment of within-cluster interactions after
cohesin degradation for one histone modification cluster in particular, cluster 4, which as we now
demonstrate in 5C, and via analysis of the pattern of histone modifications on the loci in this cluster in 5E,

corresponds to a B-type subcompartment. We have additionally clarified these points in the text and in the
legend of Fig. 5 (MS lines 354-403)

6. p.15, Fig. 6A and 6C: I did not understand the meaning of the '-auxin' data, i.e. the left-bottom heatmap
in both Fig. 6A and 6C. The same is true for cluster 4 in Fig. 6C and 6D. Also, what do the red square
numbers (legend embedded within Fig. 6C) mean? Why does the Hi-C profile of the '+auxin (28h)' cells
appear low resolution in Fig. 6C? Please explain.

We thank the reviewer for these helpful requests for clarification. For both 5A and 5C (previously 6A and
6C), two versions of the untreated (-auxin) data are shown, because they are matched controls (harvested
at the same time) for the two treated timepoints respectively, i.e. the top left heatmap in both 5A/C
correspond to the matched control for the 6hr auxin timepoint, and the bottom left heatmap in both 5A/C
corresponds to the matched control for the 28hr auxin timepoint. This has now been clarified in the Figure
legend. Cluster 4 corresponds to particular set of loci identified on the basis of clustering of histone
modification patterns that demonstrate particularly heightened compartment interactions upon cohesin
degradation. We apologize for the confusion, and have now reworded the figure caption to make this point
clearer. The red square legends correspond to the maximum intensity of the contact heatmaps shown.
The minimum intensity is 0. This has now been clarified in the figure legend. The 28hr auxin treatment
timepoint maps were not sequenced to loop resolution, these maps are lower resolution than the 6hr auxin
treatment maps. However, as the focus of this study is on the behavior of compartment features much
larger than loops, we do not believe that the lower resolution of the 28hr auxin treatment maps confounds
the conclusions presented in this manuscript (MS lines 354-403)

7. Fig. 6B: the '+auxin (28h)' cells must include a significant number of non-MLN cells (~40%). How can
the authors claim that postmitotic MLNs reconstituted the A/B compartments? (p. 14, line 321)

561 A: Most Hi-C data are averaged over large cell numbers (> 1 million) and it is commonly accepted that
they comprise heterogenous cell populations (e.g. a fraction of ~ 30% of mitotic cells in previous Hi-C
studies obtained from cells after 6h cohesin depletion). It should also be noted that MLN originating from
endomitosis have a chromosome content between 2C and 4C (after S phase of these endomitotic nuclei),
which increases their relative DNA fraction up to an estimated value of ~80% of the entire cell population.
This fraction can be considered as sufficient for shaping the predominant pattern, supported by our finding
that compartmentalization (restoring A/B compartments) is even strengthened. It is also important to note
that a large proportion of the non-MLN cells at the 28hr treatment timepoint correspond to cells still
passing through mitosis, and it is well established that mitotic cells do not exhibit A/B
compartmentalization. Thus, the strengthening of A/B compartmentalization at the 28hr auxin treatment
timepoint observed in our data is most parsimoniously explained by the reconstitution of A/B
compartments in postmitotic MLNs.

8. Fig. 8C: The claim is unclear: the authors need to explain more about this figure. The same can be said
about the main text (p. 18, line 388–393): the description is clearly inadequate. At present, I don't see
anything convincing that supports the authors' conclusion that there is strong correspondence between
replication timing and genome compartmentalization with A and B compartments before and after cohesin
depletion. Also, what are the arrows in Fig. 8C?

579 A: We apologize for the accidentally presented arrows and removed them. In line with a revision of former
Fig. 8, now revised Fig. 7, we revised the text of this paragraph (MS lines 442ff). Panels A and B in this
revised Figure support the conclusion that global replication timing noted in controls (without auxin) is
maintained in auxin-treated cells as shown in (A) by the correspondence of the untreated and auxin-
treated Repli-Seq tracks for chromosome 8 together with contact matrices of chromosome 8 at 500 kb
resolution and in (B) by the scatter plot of replication timing (percentile of E/L ratio). The new panel C
supports the tight relationship between genome A/B compartmentalization and replication timing, which
was maintained in the absence of cohesin. The top part presents the overlay of nearly identical tracks of
replication timing data for human chromosome 3 ($\log_2(E/L)RT$) from cells cultured with auxin for 6 h (blue)
and control cells without auxin (black). The bottom part shows the result of a principle component 1
analysis of Hi-C data for chromosome 3 (PC1, also known as Eigenvector). Data were based on at least
two replicates of each timepoint.

9. Fig. 9A and 9B: The number of foci in the '30h auxin' nuclei (Fig. 9A) looks smallest among the three
images by visual inspection but Fig. 9B says otherwise. Also, can you zoom in on the individual foci in

Fig.9? I say this because at this resolution, it is difficult to see whether these individual foci really
expanded in volume as the bar plot in Fig. 9C suggests.

596 A: We have added inset magnifications from each nuclear section (now Fig. 8A). Yet, a single 2D nuclear
section cannot fully reflect the situation of an entire nuclear volume. As to the „matching“ the visual
impression of Fig. 8A with quantitative data in Fig. 8 B-C: We agree that 2D size differences of RDs
between controls and cohesin depleted nuclei may not appear impressive, but even slightly increased
diameters have a remarkable impact on volumes, taking into consideration that volumes increase with the
3rd power of the radius. Our data shown in Fig. 8B-C are based on quantitative analyses from entire
image stacks and present the number of RDs counted in each studied nucleus separately (instead of
summarized RD counts for each series of nuclei in the old Figure). In Fig. 8C we provide volume
measurements for individual RDs (instead of the mean total volume of RDs/nucleus in each series). For a
detailed answer please refer to our respective answers to reviewer #2 and #3 (PBP lines 242ff and 414ff)

Minor points:

p. 2, line 45–48: long and wordy sentence

610 A: see revised abstract with shorter sentences and concise wording.

p. 4, line 134: what do you mean by "two structures"?

613 A: sentence has been deleted in the revised manuscript

p. 4, line 139: I didn't understand the following sentence: "However, we find that the physical size of
replication foci is smaller." Maybe the authors meant "larger in cohesin-depleted cells?"

617 A: We thank the reviewer for noticing this error.

p. 5, line 162: life > live

done

p. 6, line 175: seemingly > seeming

done

p. 9, Fig.2C and 2D, the authors should show the ratio of abnormal mitosis.

626 A: A previous paper of *Diaz-Martinez et al. (2010)* describes ~50% of multipolar mitoses in live cell
observations recorded within 3h after onset of mitosis. After viewing a large number of mitotic events in
live cells or para-formaldehyde fixed cells in situ (for further examples see
<https://cloud.bio.lmu.de/index.php/s/rZxxkgYExonWLGy?path=%2F>, Fig. 2 and 3), we decided against an
attempt to provide a ratio of abnormal mitoses in control and cohesin-depleted cell cultures. At face value,
cohesin-depleted cells seem to enter mitosis without a noticeable delay (see above PbP lines 140ff) and
are able to proceed to anaphase. Most of the extended period of time, which cohesin-depleted cells need
to finish endomitosis is apparently spent in failed attempts to complete karyokinesis with subsequent
cytokinesis. A more detailed classification would require the identification of stage-specific markers, which
was beyond the scope of the present study (see PbP, lines 156-167).

p. 17, Fig. 7D: the legend says 'mid to late' but it only contains 'early' and 'mid' foci patterns. Actually, the
figure format makes it very difficult for the readers to delineate Fig. 7C, 7D, and 7E.

639 A: We revised the Figure, which is now Fig. 6. The legend was corrected accordingly. To make it easier
for readers, we re-organized Fig. 6 into clearly separated blocks A-E and added brief experimental
schemes on top of each block.

p. 19, line 400: replication and timing -> replication timing

done

p. 21, line 442: What is the conclusion of the DNA halo assay? The authors could add a sentence at the
end of the last paragraph of the Results section.

648 A: We added a brief conclusion at the end of the respective result section (MS lines 507ff).

**Reviewer #4 (Remarks to the Author):**

*This manuscript studies the long-term effects of cohesin depletion on human colon cancer derived cell line*
*HCT116RAD21-mAC. Cohesin depletion is achieved by addition of auxin, which in this cell line results in*
*disintegration of cohesin from chromatin. The long term effects are delayed mitosis and multilobulated*
*nucleus (MLN). Various experiments were performed by live-cell spinning disk confocal and fixed cell SIM*
*super-resolution microscopy, as well as Hi-C and Repli-Seq, comparing control cells that underwent*
*normal mitosis with the MLN cells, and find that despite cohesin depletion there are no significant*
*differences in microscopic organisation of nuclear material. For me it is not immediately clear what is novel*
*in this study and what the information gained in this study is applicable to. Why are the MLNs the focus of*
*this study, since these do not progress through mitosis normally? The title states “Cohesin depleted cells*
*rebuild active and inactive nuclear compartments after mitosis” but why is this important? Would be helpful*
*if this was stated clearly in the introduction.*

*A: To better pointing out the novelty of our present study in the context of previous cohesin studies, we*
*substantially revised the entire manuscript. Oldach and Nieduszynski (2019) showed that cohesin ablation*
*does not disrupt patterns of replication timing investigated by RepliSeq) or BrdU-replication labeling in*
*cells fixed prior to their entry into mitosis. The present study confirms this finding. The study of Oldach and*
*Nieduszynski, however, did not include cohesin-depleted cells, which proceed through mitosis, but lack the*
*ability to complete karyokinesis with the outcome of post-endomitotic MLN cells. MLN cells were*
*chosen as the focus of this study because of their ability to re-build basic features of the*
*compartmentalized functional nuclear architecture described by the ANC-INC model. MLN show the same*
*hallmarks of active and inactive nuclear compartments, detected before in numerous cycling cell types*
*both by microscopic studies (termed ANC/INC (Cremer et al., 2020; Cremer et al., 2015) and by Hi-C*
*analysis (termed A/B compartments (Dixon et al, 2012; Lieberman-Aiden, 2009). The functional capacity*
*of the compartmentalized architecture of MLN was demonstrated by their ability to pass through a new*
*round of S-phase with typical stage-specific patterns of replications domains RDs) despite the continuous*
*absence of cohesin. These findings were not expected in view of the fundamental functional roles*
*ascribed to cohesin for the formation of chromatin loop and TADs (see Introduction). Furthermore, our*
*study indicates that cohesin in addition to a normal course of mitosis is required for the full structural*
*maintenance of RDs. In comparison with control nuclei, we noted an increase of RD counts in cohesin-*
*depleted cells grown for 6h in auxin after replication labeling post-endomitotic MLN grown for 30h in auxin*
*after RL (rev. Fig. 8). The current study so far provides the most detailed comparison of quantitative 3D*
*image analyses of cohesin-depleted post-endomitotic MLN with nuclei of pre-mitotic cohesin-depleted*
*cells and control nuclei with concurrent Hi-C analyses.” (see above, PbP letter lines 14-20).*

*The discussion mentions some situations where MLN occur but leaves it unclear whether these results are*
*relevant to these conditions and, for example, treatment of any disease.*

*A: The morphology of MLN is not unique, but resembles multilobulated nuclei e.g. in megakaryocytes*
*Whereas such multilobulated nuclei apparently serve a physiological role, MLN after cohesin depletion*
*cells likely represent an end-stage. We did not observe that MLN cells would enter another mitosis. The*
*focus (and novelty) of our study is on the ability of MLN cells to rebuild a functional nuclear architecture in*
*an aberrant daughter nucleus. Mechanisms leading to multilobulation are complex and have remained*
*speculative in cohesin depleted cells. Their further disclosure was beyond the scope of our study. Detailed*
*structural and functional comparisons of MLN with other multilobulated cells were beyond the scope of the*
*present study. This issue is discussed in MS lines 532ff*

*None of the microscopy and analysis methods are new. The authors state that “With super-resolved*
*microscopy we demonstrate that nuclei of pre- and postmitotic cohesin depleted cells maintain principal*
*structural features of the ANC-INC model” - this has been shown to be true for different cell types in refs*
*14, 15 using the exactly same microscopy and analysis methods as in this paper. While these studies do*
*not look at cohesin depletion, ref 14 states “Notably, these principal structural features of the ANC-INC*
*model are also maintained in cohesin depleted nuclei [93] [94] and even re-constituted in these cells after*
*mitosis [93]”. Therefore I am not sure what is the novelty of the current study?*

*A: References 14 (Cremer et al., 2020) and 15 (Cremer et al., 2015) review work of many years which laid*
*the fundamentals for the present study. Essential features of the ANC-INC model based on studies of cell*
*types from a variety of mammalian species described in our previous publications do in no way predict the*
*outcome of the present study with its intention to explore to which extent cohesin may be indispensable or*
*turn out to be dispensable for different features of the ANC-INC model. Ref [93] in Cremer et al. 2020*

refers to the preprint, which we posted on *BioRxiv* on October 2019 <https://doi.org/10.1101/816611>. It
corresponds to the manuscript currently under review here in NCOMMS. Ref [94] in *Cremer et al. 2020*
refers to the preprint “Chromatin arranges in filaments of blobs with nanoscale functional zonation posted
on *biorXiv* on March 16, 2019 from E. Miron, ... L. Schermelleh, (<https://doi.org/10.1101/566638>). The
authors concluded: “Our findings support a model of a higher-order chromatin architecture on the size
level of TADs that creates and modulates distinct functional environments through a combination of
biophysical parameters such as density and ATP-driven processes such as replication and transcription,
but independent of cohesin.” On May 30, 2020 these authors posted a revised version on *BioRxiv*
(<https://doi.org/10.1101/566638>) “Chromatin arranges in chains of mesoscale domains with nanoscale
functional topography independent of cohesin”:
“High-content mapping uncovers confinement of cohesin and active histone modifications to surfaces and
enrichment of repressive modifications towards the core of CDs in both hetero- and euchromatic
regions. This nanoscale functional topography is temporarily relaxed in post-replicative chromatin, but
remarkably persists after ablation of cohesin. Our findings establish CDs as physical and functional
modules of mesoscale genome organization.”
Like ours, this paper is apparently still under review. Our study and the Miron et al. study complement and
strengthen each other, but none is published to date in a peer-reviewed journal.

*The authors use 3D SIM to measure chromatin compaction of nuclei. This is done by classification of*
*DAPI intensity in the 3D-SIM images in seven classes. This method is described in ref 22 and has been*
*used in other studies e.g. 14, 15. The results in Fig 3 show similar topography of chromatin compaction in*
*control and cohesin depleted nuclei. Would this not be expected?*

729 A: All previous 3D SIM studies were performed with cells expressing the proteins required for the
730 formation of cohesin rings and a higher order chromatin organization in the presence of such rings. The
731 starting point of the present study was our observation (*Rao et al., 2017*) that cohesin-depleted cells
studied with ensemble Hi-C demonstrated the loss of triangles reflecting TADs. This finding argued for a
loss of all cohesin-dependent chromatin loops. The time window studied in *Rao et al., 2017* included only
cells within one cell cycle, not post-endomitotic MLN described here. Therefore, this previous study does
not bear on the outcome of the endomitotic cell cycle of a cohesin-depleted cell.

*In Fig 4 the authors aim to demonstrate IC-channels extending between clusters of lamina associated*
*domains into the nuclear interior. These images do not have sufficient resolution to make this conclusion.*
*Figs 4a-c appear to show some vertical channels but it is also possible that these images contain some*
*artefacts as these vertical stripes are very regular and prominent. Why are these channels always vertical,*
*including at the ends of the nucleus where I would expect to see horizontal channels towards the centre?*
*Figs 4d-f show areas from a-c with intensity classification which does show some of these channels (in*
*darker areas of the images) extending to the centre, but in brighter areas these are missing after the*
*intensity classification, further indicating that the intensity variations could be caused by other sources e.g.*
*image reconstruction.*

746 A: Fig. 4 (now shown as Supplemental Fig. 8) suggests a predominance of vertically running IC channels.
This predominance of a vertical channel alignment in xz or yz sections is an optical effect. Compared to xy
sections (perpendicular to the optical axis with a resolution of ~125 nm (*Schermelleh et al., 2010*),
resolution along the z-axis is considerably lower (~300 nm), prompting the apparent (but not real!) loss of
horizontal channels. In the revised Figure, we added an xy section, which demonstrates channels
extending through the peripheral layer of compact chromatin in a fairly horizontal direction to demonstrate
that the channel system is a three-dimensional network. We have clarified this in the detailed legend of
supplementary Fig. 8.

As to the comments regarding previous Fig. 4D-F (now suppl. Fig 8G-I): In a previous study (*Schmid et*
*al., 2017*), we showed that use of an increased number of DAPI intensity classes beyond the routinely
chosen 7 classes did not affect our conclusions. A 3D image analysis of nuclei based on individual voxels
reflecting different DNA densities is possible but increases the computational load. Direct connections of
IC-channels to nuclear pores were described in previous studies (*Schermelleh et al., 2010; Smeets et al.,*
*2014*); for a detailed review of the IC and its potential participation in the functional organization of the cell
nucleus see *Cremer et al., 2020*.

*In Fig 5 it is difficult to see where exactly the SC35, H3K27me3 and active RNA Pol II are located. For*
*example, the authors state “RNA Pol II was enriched in the PR, i.e. decondensed chromatin lining the IC”*

*but it is very difficult to assess whether this is true from the images that are shown. I would suggest*
*removing DAPI from the inset images left images (while showing the DAPI classified image next to it) to*
*make this clearer.*

*A: Our statement “RNA Pol II was enriched in the PR, i.e. decondensed chromatin lining the IC” was*
*based on a quantitative 3D evaluation of entire nuclei. We agree that images should be typical and*
*exemplify respective statements, but single optical sections cannot reflect an entire nucleus and there is*
*no “one fit for all” nucleus. We think however, that our presentation with inset magnifications of nuclear*
*markers on DAPI (grey) put side by side with DAPI shown as intensity correlated heat map as proxy for*
*chromatin compaction provides an appropriate way of presentation. As for the variability between nuclei*
*and interexperimental differences we refer to the new Supplementary Fig. 6 and MS lines 309-350.*

*In Fig 9 please show enlarged example areas of segmentation.*

*A: We added inset magnifications for each nucleus. For further comments regarding the data of this Figure*
*(now Fig. 8) we refer to our respective answers to reviewers #2 (PbP lines 252ff) and to reviewer #3 (PbP*
*lines 434ff and 604ff)*

REVIEWERS' COMMENTS

Reviewer #3 (Remarks to the Author):

The authors have addressed most of my concerns and the manuscript reads much better now. I like Supplementary Table 1, which will be very helpful for the readers in and out of the 4D nucleome field. I just have a few comments that could further improve the manuscript.

(1) Panel C of revised Figure 7 described in the rebuttal letter seems to be missing.

Please see the second half of the authors' comments below. I couldn't find chromosome 3 anywhere so I believe the figure is missing.

A: We apologize for the accidentally presented arrows and removed them. In line with a revision of former Fig. 8, now revised Fig. 7, we revised the text of this paragraph (MS lines 442ff). Panels A and B in this revised Figure support the conclusion that global replication timing noted in controls (without auxin) is maintained in auxin-treated cells as shown in (A) by the correspondence of the untreated and auxin treated Repli-Seq tracks for chromosome 8 together with contact matrices of chromosome 8 at 500 kb resolution and in (B) by the scatter plot of replication timing (percentile of E/L ratio).

The new panel C supports the tight relationship between genome A/B compartmentalization and replication timing, which was maintained in the absence of cohesin. The top part presents the overlay of nearly identical tracks of replication timing data for human chromosome 3 ($\log_2(E/L)RT$) from cells cultured with auxin for 6 h (blue) and control cells without auxin (black). The bottom part shows the result of a principle component 1 analysis of Hi-C data for chromosome 3 (PC1, also known as Eigenvector). Data were based on at least two replicates of each timepoint.

(2) Relationship between RDs, compartment domains and TADs

Nothing wrong with having one's own view, but the Pope et al. paper (ref. 49) by one of the co-authors concluded that RDs correspond to TADs. Do all the authors agree on the view that RDs correspond to compartment domains?

Also, as I said in the initial review, A/B compartment domains are usually larger than 1 Mb and sometimes can be up to or even over 10 Mb. I think the authors should be cautious and discuss whether RDs or ~1 Mb chromatin domains (replication foci) can be over 10 Mb.

In this sense, the authors might want to refer to the recent single-cell Repli-seq papers, as they are relevant for the discussion on lines 551–581. See for instance: Dileep et al., Nat Commun (2018); Takahashi et al., Nat Genet (2019); Hiratani and Takahashi, Genes (2019); Donaldson and Nieduszynski, Genome Biol (2019); Miura et al., Nat Genet (2019).

(3) Line 613-615: "In summary, we tentatively conclude that cohesin plays an indispensable role for the structure of RDs/CDs but is dispensable for the formation of a compartmentalized nuclear organization."

In this summary sentence near the end of the Discussion section, the authors should talk about MLN and what they learned new from it. Without it, the conclusion is identical to the cohesin degron paper (Rao et al., Cell 2017).

Minor points:

p.2, line 52: which correspond to A/B compartments, > which correspond to A and B compartments, respectively,

p.23, line 558: several 100 kb > several hundred kb

p.25, line 589: 6 and 30 h respectively resulted in > 6 and 30 h revealed

Reviewer #5 (Remarks to the Author):

The authors have addressed all my comments in a satisfactory manner their reply and the revised manuscript.

Additionally, revision of the manuscript text has greatly improved readability and made the aims, context and conclusions of the study easier to understand.

I recommend the manuscript for publication in its current form.

Final comments on the remaining concerns of Reviewer #3

Comment: The authors have addressed most of my concerns and the manuscript reads much better now. I like Supplementary Table 1, which will be very helpful for the readers in and out of the 4D nucleome field. I just have a few comments that could further improve the manuscript.

Answer: We thank reviewer #3 for her/his positive statement about the improvements of the revised version, including Supplementary Table 1, and again for his persistence in emphasizing major, unresolved problems of the basic structural features of the nuclear landscape and their implications for nuclear functions (see below).

Comment: (1) Panel C of revised Figure 7 described in the rebuttal letter seems to be missing. Please see the second half of the authors' comments below. I couldn't find chromosome 3 anywhere so I believe the figure is missing.

Answer: After a thorough discussion, we decided to include Fig. 7A,B in our manuscript revision. We apologize that by mistake, we still referred to the now deleted Fig. 7C in our previous point-by-point answers. The revised Figure 7 provides evidence that replication timing along the genome is preserved in cohesin depleted cells 6 h after auxin treatment. This conclusion is in line with published data. As compared with Fig. 7A in our original submission, we added the Eigenvector in the revised Fig.7A. Additional data in the deleted Fig. 7C for another region studied on chromosome 1 would not substantially add to this conclusion.

Comment: (2) Relationship between RDs, compartment domains and TADs. Nothing wrong with having one's own view, but the Pope et al. paper (ref. 49) by one of the co-authors concluded that RDs correspond to TADs. Do all the authors agree on the view that RDs correspond to compartment domains? Also, as I said in the initial review, A/B compartment domains are usually larger than 1 Mb and sometimes can be up to or even over 10 Mb. I think the authors should be cautious and discuss whether RDs or ~1 Mb chromatin domains (replication foci) can be over 10 Mb. In this sense, the authors might want to refer to the recent single-cell Repli-seq papers, as they are relevant for the discussion on lines 551-581. See for instance: Dileep et al., Nat Commun (2018); Takahashi et al., Nat Genet (2019); Hiratani and Takahashi, Genes (2019); Donaldson and Nieduszynski, Genome Biol (2019); Miura et al., Nat Genet (2019).

Answer: We share the concerns of reviewer #3 concerning very different, even conflicting points of view with regard to structural and functional relationships between entities that contribute to the higher order chromatin organization above the nucleosome level. Relationships between RDs, TADs or compartment domains represent a case in point. We suggest to equate RDs with CDs and compartment domains rather than with TADs based on evidence that RDs, CDs and compartment domains were detected in both premitotic and postendomitotic cohesin depleted nuclei, whereas we missed TADs in ensemble Hi-C experiments. This view is, however, challenged by the observation of TAD-like domains in experiments that combined oligo-paint FISH of sequence-defined TADs with super-resolved fluorescence microscopy. These experiments indicate that the failure to detect TADs in cohesin depleted cells in ensemble Hi-C experiments may not reflect a real loss of structural entities, but depends on apparently random cell-to-cell shifts of boundaries, demarcating neighboring TADs.

To emphasize the current difficulties to achieve a commonly accepted terminology, we present separated glossaries for terms based on microscopic studies (Suppl. Table 1A) and for terms based on Hi-C and biochemical evidence (Suppl. Table 1B). In 1B, we incorporated two single-cell Repli-seq papers mentioned by the reviewer: Takahashi et al., Nat Genet (2019); Miura et al., Nat Genet (2019). We distinguish between compartment domains A and B, and compartments A and B. In this terminology compartment A comprises all

compartment domains A and compartment B all compartment domains B. Notably, the size of a given compartment domain depends on resolution. The observation of a compartment domain with a size of 10 MB at lower resolution does not answer the question, whether and how such a 10 MB domain may be composed of smaller subunits, when studied at higher resolution up to the level of individual nucleosomes. We admit the same uncertainty with respect to CDs, RDs and TADs. All these structures (in our current opinion) typically comprise a few hundred kb up to 1 Mb in human cell nuclei studied to date. Some of them may be smaller, others larger. After much debate, we prefer to subsume compartmental domains independent of their small size as a type of compartment domains, but leave the issue unsettled, whether a clear distinction between such small compartmental domains, which may even comprise only a single transcription unit, and much larger compartment domains should be preferred. Such examples do not invalidate the current concept of a hierarchically defined structural and functional higher order chromatin organization, but necessitate a reconsideration of this concept. A wide size range is suggested by evidence for a dynamic behavior of these structures at the single cell level. While we consider some structural and functional overlaps, the extent of such an overlap is not known.

These glossaries are followed by a comment, which argues that a common terminology should be based on clearly defined experimental approaches. The history of the term chromosome can serve as a case in point. When Wilhelm Waldeyer (1836-1921) introduced this term in 1882, he was aware of August Weismann's (1834-1914) ingenious, but highly speculative theory of heredity. He even referred to Johann Friedrich Miescher's (1844-1895) discovery of "nuclein" and to Albrecht Kossel's (1853-1927) early publications on "histon" and "adenin" (references included in revised Supplemental Table 1). Waldeyer, however, preferred to propose the name chromosomes to emphasize the possibility of coloring the worm-like entities, seen during mitosis, by certain stains. This term has remained open for all conceptual changes, which happened thereafter with regard to the structure and function of chromosomes to the present day.

In Suppl. Table 1C we present our own preliminary attempt to integrate the different perspectives of microscopy and Hi-C and tentatively suggest hypotheses, whose validity requires further experimental tests. By dividing the revised Suppl. Table 1 into three parts, we have tried to clearly separate the presentation of terminology in the scientific literature (A and B) from our own hypotheses (C).

Comment: (3) Line 613-615: "In summary, we tentatively conclude that cohesin plays an indispensable role for the structure of RDs/CDs but is dispensable for the formation of a compartmentalized nuclear organization." In this summary sentence near the end of the Discussion section, the authors should talk about MLN and what they learned new from it. Without it, the conclusion is identical to the cohesin degron paper (Rao et al., Cell 2017).

Answer: We have changed the text accordingly. In summary, the current study supports our previous conclusion (Rao et al, 2017) that cohesin plays an indispensable role for the structure of RDs/CDs, but is dispensable for the formation of a compartmentalized nuclear organization with preserved A and B compartments. These earlier results are substantially enhanced through the microscopic observations described in the present study, which demonstrates that cohesin depleted cells passing through an endomitosis are able to rebuild a cell nucleus with the basic features of the ANC and INC, respectively. It is of note to emphasize here that Hi-C alone did neither allow to detect the drastic changes of the global architecture between cohesin depleted cells studied before and after endomitosis nor the added topographical features of IC-channels and lacunas.

Comment: Minor points:

p.2, line 52: which correspond to A/B compartments, which correspond to A and B compartments, respectively,
p.23, line 558: several 100 kb > several hundred kb
p.25, line 589: 6 and 30 h respectively resulted in > 6 and 30 h revealed

Answer: We incorporated these minor points into the final submission